# A median fin derived from the lateral plate mesoderm and the origin of paired fins

Keh-Weei Tzung[1,2], Robert L. Lalonde[3], Karin D. Prummel[3,4], Harsha Mahabaleshwar[2], Hannah R. Moran[3], Jan Stundl[5,6], Amanda N. Cass[7], Yao Le[8], Robert Lea[9], Karel Dorey[9], Monika J. Tomecka[1], Changqing Zhang[2], Eline C. Brombacher[4], William T. White[10], Henry H. Roehl[11], Frank J. Tulenko[12], Christoph Winkler[8], Peter D. Currie[12,13], Enrique Amaya[14], Marcus C. Davis[15], Marianne E. Bronner[5], Christian Mosimann[3,4 ✉] & Tom J. Carney[1,2 ✉]

The development of paired appendages was a key innovation during evolution and facilitated the aquatic to terrestrial transition of vertebrates. Largely derived from the lateral plate mesoderm (LPM), one hypothesis for the evolution of paired fins invokes derivation from unpaired median fins via a pair of lateral fin folds located between pectoral and pelvic fin territories[1]. Whilst unpaired and paired fins exhibit similar structural and molecular characteristics, no definitive evidence exists for paired lateral fin folds in larvae or adults of any extant or extinct species. As unpaired fin core components are regarded as exclusively derived from paraxial mesoderm, any transition presumes both co-option of a fin developmental programme to the LPM and bilateral duplication[2]. Here, we identify that the larval zebrafish unpaired pre-anal fin fold (PAFF) is derived from the LPM and thus may represent a developmental intermediate between median and paired fins. We trace the contribution of LPM to the PAFF in both cyclostomes and gnathostomes, supporting the notion that this is an ancient trait of vertebrates. Finally, we observe that the PAFF can be bifurcated by increasing bone morphogenetic protein signalling, generating LPM-derived paired fin folds. Our work provides evidence that lateral fin folds may have existed as embryonic anlage for elaboration to paired fins.

Two alternate hypotheses have been proposed to explain the evolutionary origin of vertebrate paired appendages (fins and limbs). Derivation from posterior gill arches was posited by Gegenbaur[3], whilst a number of anatomists later invoked a rival theory, the lateral fin fold hypothesis. This proposed that paired fins derived (either phylogenetically or ontogenetically) from longitudinal bilateral fin folds that were then subdivided[4,5]. Whilst recent molecular studies have provided some evidence in support of each hypothesis[2,6,7], there remains significant criticism of the lack of substantiation in the fossil record or in embryology[1,8]. Certain stem vertebrates, including anaspid-related fossils, show evidence of lateral fin folds; however, these fin folds mostly consist of soft tissue with only sporadic skeletal elements and are thus poorly preserved. This has led to conflicting interpretations[9–13]. The developmental programme for paired fins has been postulated to have been first assembled in median fins, which appear in the fossil record before the origin of paired fins[2]. A number of studies have traced the cellular origin of median fins in lamprey,

catshark, zebrafish and *Xenopus*[2,14,15]. All median fins, both larval and adult, assessed so far have shown derivation from the paraxial mesoderm (PM), whilst paired fins are known to be derived from the lateral plate mesoderm (LPM). Thus, the median fin programme was most likely transferred to the LPM from the PM[2], possibly before the formation of hypothesized lateral fin folds. How or when this transition occurred is unclear. As only a subset of median fins in zebrafish has been assayed, we expanded the characterization of the composition and origin of median fins in zebrafish to determine if PM derivation was an invariant characteristic.

As with most surveyed jawed vertebrates (gnathostomes), larval zebrafish possess two median unpaired fin folds. A caudal fin fold (or major lobe) runs continuously from the dorsal midline around the caudal end of the larva and then ventrally to the anus (Fig. 1a). A pre-anal fin fold (PAFF; or minor lobe) runs along the underside of the yolk sac extension, immediately anterior to the anus[16] (Fig. 1a). The median fin folds are resorbed during metamorphosis, and the caudal fin fold is

[1]Institute of Molecular and Cell Biology, A*STAR, Singapore, Singapore. [2]Lee Kong Chian School of Medicine, Nanyang Technological University, Singapore, Singapore. [3]Department of Pediatrics, Section of Developmental Biology, University of Colorado Anschutz Medical Campus, Aurora, CO, USA. [4]Department of Molecular Life Sciences, University of Zurich, Zurich, Switzerland. [5]Division of Biology and Biological Engineering, California Institute of Technology, Pasadena, CA, USA. [6]Faculty of Fisheries and Protection of Waters, University of South Bohemia in Ceske Budejovice, Vodnany, Czech Republic. [7]Biology Department, Wesleyan University, Middletown, CT, USA. [8]Department of Biological Sciences, Faculty of Science, National University of Singapore, Singapore, Singapore. [9]Division of Developmental Biology and Medicine, School of Medical Sciences, Faculty of Biology, Medicine and Health, University of Manchester, Manchester, UK. [10]CSIRO National Research Collections Australia, Australia National Fish Collection, Hobart, Tasmania, Australia. [11]School of Biosciences, University of Sheffield, Sheffield, UK. [12]Australian Regenerative Medicine Institute, Monash University, Clayton, Victoria, Australia. [13]EMBL Australia, Victorian Node, Monash University, Clayton, Victoria, Australia. [14]Division of Cell Matrix Biology and Regenerative Medicine, School of Biological Sciences, Faculty of Biology, Medicine and Health, University of Manchester, Manchester, UK. [15]Department of Physical and Biological Sciences, Western New England University, Springfield, MA, USA. ✉e-mail: christian.mosimann@cuanschutz.edu; tcarney@ntu.edu.sg

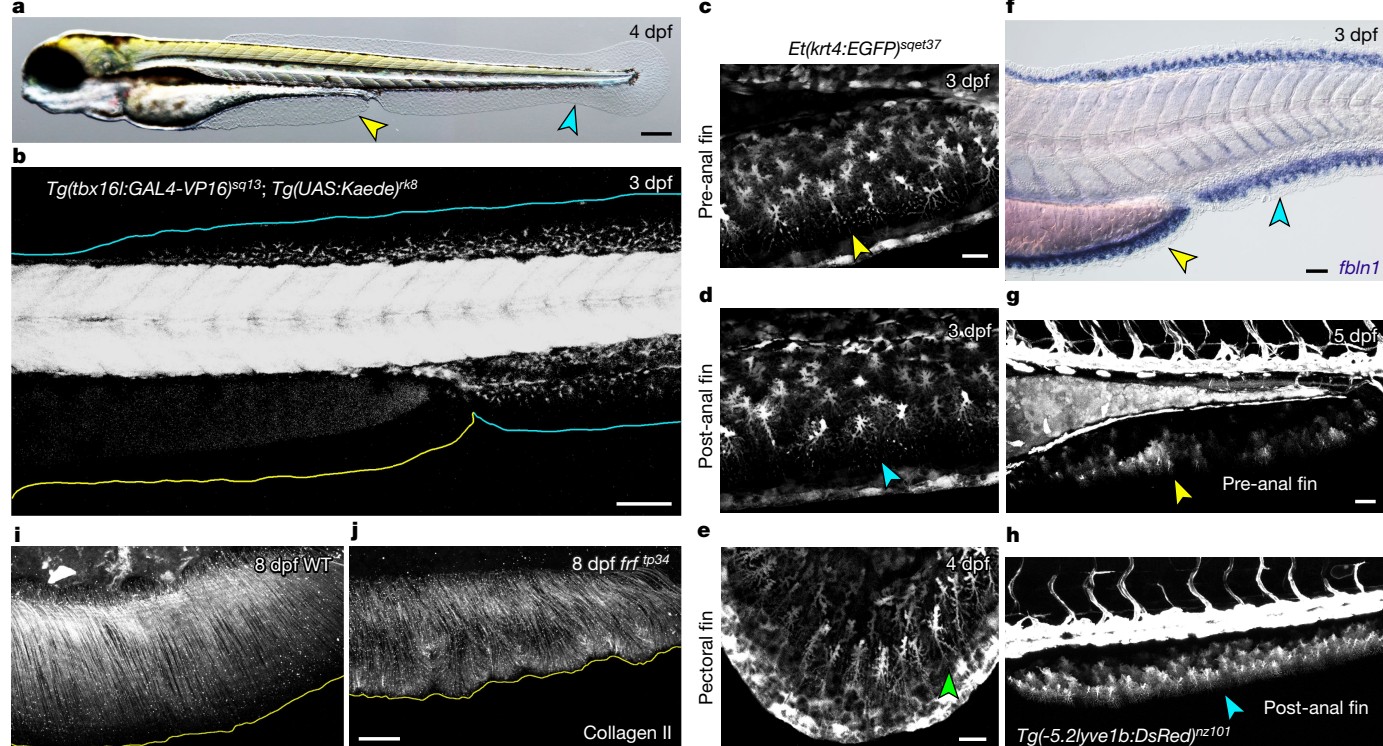

**Fig. 1 | A non-PM-derived median fin fold. a**, Larval 4 dpf zebrafish possess a median PAFF (yellow arrowhead) in addition to the caudal median fin fold (cyan arrowhead). **b**, Confocal image of a 3 dpf *Tg(tbx16l:GAL4-VP16); Tg(UAS:Kaede)* embryo with PM labelled by Kaede showing PM-derived mesenchyme in the caudal median fin fold (cyan outline) but not the PAFF (yellow outline). **c–e**, Confocal images of pre-anal (**c**), ventral caudal (**d**) and pectoral (**e**) fins of the *ET37* Enhancer Trap transgenic line indicating that PAFF contains morphologically comparable mesenchyme (indicated by arrowheads) to other larval fin folds. **f**, In situ hybridization of the fin mesenchyme marker *fbln1* in both PAFF (yellow arrowhead) and caudal fin fold (cyan arrowhead) at 3 dpf. **g,h**, DsRed expression in mesenchyme of both pre-anal (**g**) and caudal (**h**) fin folds of the 5 dpf *Tg(-5.2lyve1b:DsRed)* transgenic line. **i,j**, Immunostaining for collagen II in 8 dpf *frf* mutants (**j**) shows loss of fibril organization compared with WT (**i**). Scale bars, 200 μm (**a**); 100 μm (**b**); 20 μm (**c,e**); 50 μm (**f,g,j**).

replaced by three separate PM-derived median adult fins[14,17]: the caudal fin, the dorsal fin and the anal fin. No adult median fin replaces the PAFF, which is a developmentally transient structure[16]. The mesenchyme of the caudal fin fold can be labelled by the photoconvertible Kaede green-to-red fluorescent protein expressed under control of the *tbx16l* promoter, confirming that this median fin is derived from the PM[18]. In contrast, we consistently failed to observe any Kaede labelling in the PAFF (Fig. 1b and Extended Data Fig. 1a). This labelling difference is not due to de novo activity of the *tbx16l* promoter specifically in the caudal median fin fold, as we photoconverted Kaede in anterior somites above the PAFF at 24 h post-fertilization (hpf) and traced the photoconverted Kaede to the mesenchyme of the dorsal portion of the caudal fin fold; again, we never found labelled cells in the PAFF (*n* = 11) (Extended Data Fig. 1b–d). The absence of PAFF labelling by a PM-lineage trace is not due to absence of mesenchyme cells, as we observed an abundance of these cells by Nomarski optics at 3 days post-fertilization (dpf) and in the enhancer trap line *Et(krt4:EGFP)*[sqet37] (*ET37*), which labels all fin mesenchyme cells (Extended Data Fig. 1e,f). The *ET37* line further allowed us to visualize the morphology of the mesenchyme cells, which showed a distally polarized stellate shape, indistinguishable from the mesenchyme of all other fins (Fig. 1c–e and Extended Data Fig. 1f–h). PAFF mesenchymal cells expressed known differentiation markers of fin mesenchyme, including *fibulin1* (*fbln1*) and *integrin beta 3b* (*itgb3b*), and were weakly positive for *lyve1b* reporter activity in the *lyve1b:DsRed2* transgenic line, which also expresses in the post-anal ventral fin mesenchyme[19] (Fig. 1f–h and Extended Data Fig. 1i). Hence, despite their divergent developmental origin, PAFF mesenchyme has a similar morphology and expression profile to fin mesenchyme of the major median fin lobe.

Few functions of fin mesenchyme cells are defined. The zebrafish *frilly fin* (*frf*) mutants display ruffling of the caudal larval fin fold due to mutations in *bone morphogenetic protein 1a* (*bmp1a*), which is expressed in fin mesenchyme[20], including in the PAFF (Extended Data Fig. 1j). The overall morphology of the PAFF also displays ruffling in *frf* mutants (Fig. 1i,j). Further, immunostaining of collagen II revealed ordered parallel collagen II fibres in all median fins of the wild type (WT), while these fibres were disorganized in both the caudal median fin folds and the PAFF of *frf* mutants (Fig. 1i,j), indicating that Bmp1a in PAFF mesenchyme is also required for maturation of collagen. Collectively, these results indicate that mesenchymal cells of the PAFF show functional overlap with those of the caudal median fin fold yet have a distinct developmental origin.

The transcription factor Hand2 is expressed in several LPM progenitors, including cardiac, pharyngeal, mesothelial and pectoral fin progenitors[21,22]. Upon examining the *TgBAC(hand2:EGFP)* transgenic line that accurately recapitulates endogenous *hand2* gene expression[23], we observed the expected enhanced green fluorescent protein (eGFP) expression in larval pectoral fins at 2–3 dpf (Fig. 2a,b). Although we observed no *TgBAC(hand2:EGFP)* expression in the major unpaired larval fin lobe, the PAFF mesenchyme was strongly eGFP positive, suggesting an LPM origin (Fig. 2a,b and Extended Data Fig. 2a,b). Additionally, in situ hybridization at 3 and 5 dpf indicated that the PAFF mesenchyme, but no other median fin fold, expressed *hand2* (Fig. 2c,d and Extended Data Fig. 2c,d). If the LPM uniquely contributes to the PAFF, then we might expect that the PAFF, but no other median fin fold, would be affected in an LPM mutant. The zebrafish *hand2* mutant *hands off* (*han*[s6]) exhibits defects in LPM derivatives, including the heart, pectoral fins and mesothelium[21,22]. Consistent with an LPM origin, the

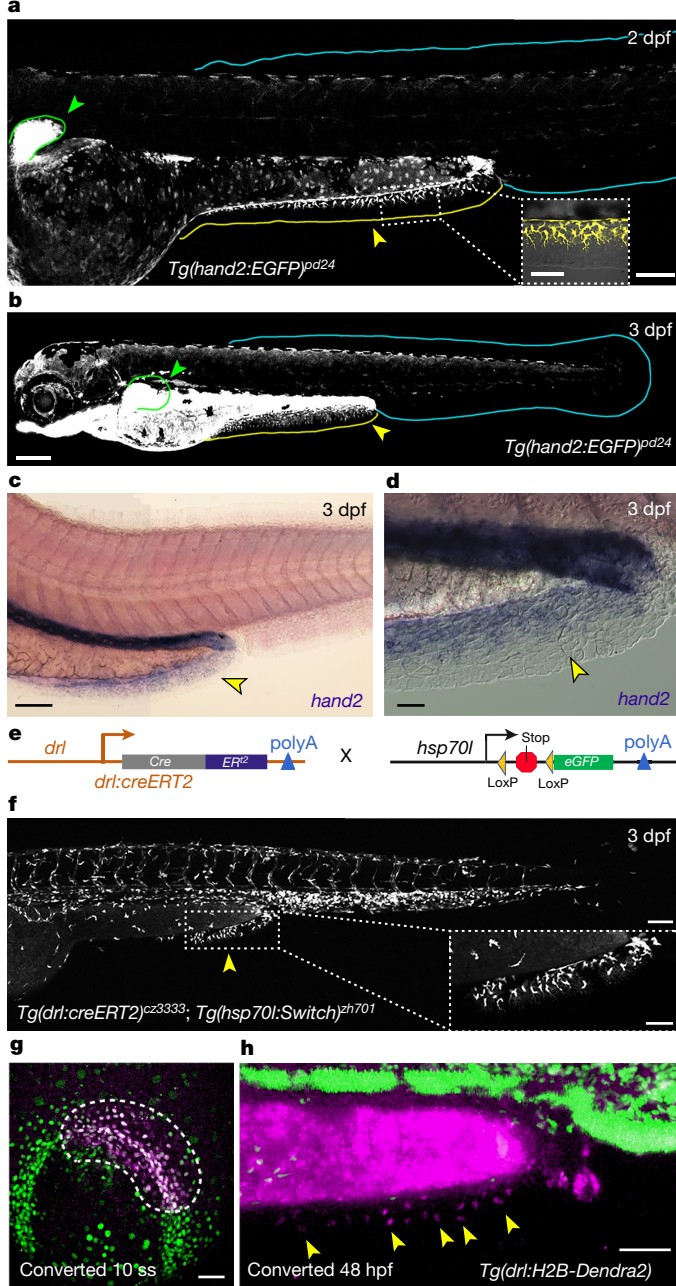

**Fig. 2 | The PAFF is an LPM-derived median fin fold. a,b**, Confocal images of *Tg(hand2:EGFP)* embryos at 2 dpf (**a**) and 3 dpf (**b**) showing eGFP labelling of the mesenchyme of the pectoral (green outline and arrowheads) and PAFFs (yellow outline and arrowheads, and magnified in inset) but not the caudal fin fold (cyan outline). **c,d**, In situ hybridization of *hand2* at 3 dpf shows fin expression of *hand2* only in the PAFF (yellow arrowheads (**c**), and higher magnification with Nomarski optics indicates expression in the mesenchyme (**d**). **e**, Schematic of the LPM lineage tracing transgenes. **f**, Lineage tracing of LPM using transgenics depicted in (**e**) following 4-OHT treatment and heat shock before imaging shows that PAFF mesenchyme is derived from the LPM (yellow arrowhead and magnified in inset). **g,h**, Ventral (**g**) and lateral (**h**) confocal images of the *drl:H2B-Dendra2* transgenic line at the 10-somite stage (10 ss) (**g**) and 48 hpf (**h**) following ultra-violet laser photoconversion in the region of the LPM outlined in (**g**). **h**, Photoconverted PAFF mesenchyme is indicated by yellow arrowheads. Scale bars, 100 μm (**a,c,f**); 50 μm (**a** (inset), **f** (inset),**g,h**); 200 μm (**b**); 20 μm (**d**).

fin height of PAFFs in *han^s6* mutant larvae was significantly reduced compared with WT at 3 and 5 dpf, whilst the ventral section of the major median fin lobe was unaffected (Extended Data Fig. 3a,b,g,h,m). We

observed similar results with *hand2* antisense morpholino (MO)-based knockdown (Extended Data Fig. 3c–f,n). *hand2* knockdown in the *ET37* line severely reduced the number of eGFP-positive mesenchymal cells in the PAFF at 3 and 5 dpf compared with WT, whilst the mesenchyme of the ventral caudal median fin fold was unaffected (Extended Data Fig. 3i–l,o). Transplantation of *hand2* MO; *ET37* mutant cells into WT suggested that the loss of Hand2 reduced clone size and altered morphology and migration of mesenchyme cells autonomously (Extended Data Fig. 4). Together, these observations are consistent with an LPM origin of PAFF mesenchyme.

To confirm that this expression of *hand2* indicates an LPM origin of the PAFF rather than de novo expression, we permanently labelled the LPM in a mosaic fashion using a *Tg(drl:creERT2; hsp70l:Switch)* transgenic combination[24] (Fig. 2e). During gastrulation and early somitogenesis, *drl:creERT2* is expressed in LPM-primed mesoderm with increasing selectivity to the LPM, suitable for lineage labelling of this mesodermal lineage using 4-OH-tamoxifen (4-OHT) induction and *loxP* reporters[24]. Consistent with an LPM origin of paired appendages, we first documented that this LPM lineage tracing approach labels cells of the paired fins, in particular the larval pectoral fins and pelvic fins. We found lineage labelling in the endoskeletal disc and mesenchyme of 4.5 dpf pectoral fins and also, the intraray fibroblasts and Zns5-positive osteoblasts of the adult pelvic fins (Extended Data Fig. 5a–d). In addition, we observed LPM lineage labelling of the PAFF mesenchyme at 3 and 5 dpf, whilst the labelling in the caudal fin fold was limited to myeloid cells, which are also LPM derived (Fig. 2f and Extended Data Fig. 5e–g). These data further support the contribution of the LPM to the PAFF mesenchyme but not to other median fin folds. We next used *drl*-driven expression of the Dendra2 green-to-red photoconvertible fluorophore to localize the precise LPM origin of PAFF mesenchyme. During early to mid-somitogenesis (8–10-somite stage (ss)), *drl:H2B-Dendra2* is expressed in the nuclei of the LPM (Extended Data Fig. 6a,b). Photoconversion of posterior-most LPM at this stage, using ultra-violet laser illumination, resulted in selective Dendra2-red labelling of this field of LPM (Fig. 2g and Extended Data Fig. 6c,d). These cells were then traced to the migrating PAFF mesenchyme at 40 and 48 hpf (*n* = 4) (Fig. 2h and Extended Data Fig. 6e–j,q–t), whilst unconverted control cells did not show any Dendra2-red PAFF labelling (Extended Data Fig. 6k–p,u–x). This confirmed PAFF mesenchyme originates from the LPM, with significant contribution from the posterior-most LPM at early segmentation stages.

To investigate whether an LPM-seeded median PAFF is a zebrafish apomorphy or if it is a more ancestral feature shared with other vertebrates, we examined the larvae of other taxa for the presence of a PAFF and used *hand2* in situ hybridization as a proxy marker of an LPM origin of constituent mesenchymal cells. Medaka (*Oryzias latipes*) larvae have a markedly smaller PAFF that possesses only a sparse number of mesenchymal cells as detected by Nomarski imaging (Fig. 3a,b). These mesenchymal cells have cell extensions projecting distally and were mostly proximal in the fin where the interstitial space is thickest (Fig. 3b). Accordingly, medaka *hand2* was expressed proximally in a punctate pattern in PAFFs, consistent with mesenchymal expression, and was not expressed in other median fin folds (Fig. 3c and Extended Data Fig. 7a–c). To confirm this, we expressed eGFP in early medaka LPM by injection of a -6.35drl:EGFP construct. We saw expression in mesenchymal cells in the PAFF transiently and also, in stable transgenic embryos (eight PAFF mesenchyme cells seen in four stage 36 *Tg(-6.35drl:EGFP)* transgenic medaka embryos) (Extended Data Fig. 7d,e). In addition, we used 1,1'-dioctadecyl-3,3,3',3'-tetramethylindocarbocyanine perchlorate (DiI) lipophilic dye labelling to label the posterior LPM territory at stage 20 (four-somite stage). Consistent with our positional mapping of PAFF-seeding mesenchyme in zebrafish, posterior LPM injection of DiI consistently labelled mesenchyme in the medaka PAFF (total of 27 cells in nine of nine labelled embryos) (Extended Data Fig. 7f–i). A basal actinopterygian, the

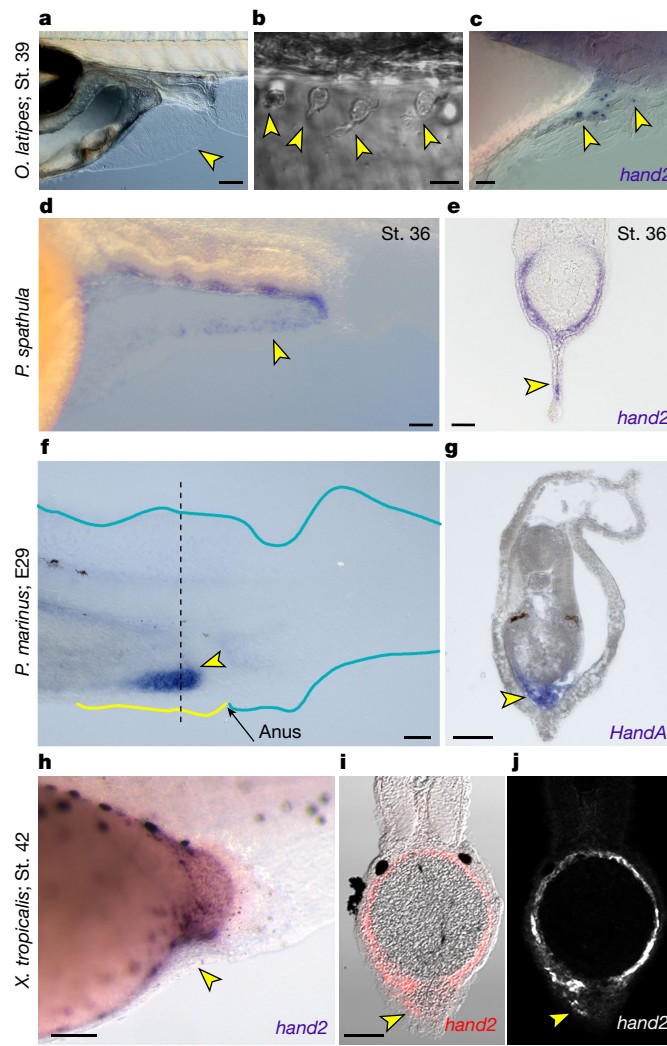

**Fig. 3 | PAFF mesenchyme expression of *hand2* is conserved across vertebrates. a**,**b**, Nomarski images of 9 dpf medaka showing PAFF (**a**) with dispersed mesenchymal cells (yellow arrowheads) (**b**). **c**, In situ hybridization of medaka at stage 39 showing *hand2* expression in pre-anal fin mesenchyme (yellow arrowheads). **d**,**e**, In situ hybridization of *hand2* in stage 36 paddlefish embryos shown laterally (**d**) or in transverse section (**e**). *hand2*-positive PAFF fin mesenchyme is indicated by yellow arrowheads. **f**,**g**, *HandA* in situ hybridization of stage E29 lamprey (*P. marinus*) embryos shown laterally (**f**) or in transverse section (**g**). Lamprey show strong expression of *HandA* in a fin anterior to the anus (yellow arrowhead in **f**) corresponding to cells in the interior of the fin (**g**). The section location of (**g**) is indicated by the dashed line in (**f**). **h**–**j**, Chromogenic (**h**) or fluorescent (**i**,**j**) in situ hybridization of *hand2* in stage 42 *X. tropicalis* embryos shown laterally (**h**) or in transverse section (**i**,**j**). Fluorescent image (**j**) is overlayed on the Nomarski image (**i**). The small PAFF in *Xenopus* contains sparse *hand2*-positive fin mesenchyme (yellow arrowheads). St., stage. Scale bars, 100 µm (**a**,**d**,**h**); 10 µm (**b**); 20 µm (**c**); 50 µm (**e**–**g**,**i**).

American paddlefish (*Polyodon spathula*), also forms a transient larval PAFF[8]; we identified the expression of *P. spathula hand2* in PAFF mesenchymal cells between stages 36 and 39, but *hand2* expression was notably absent from the caudal median fin fold (Fig. 3d,e and Extended Data Fig. 7j–l). PAFF *hand2* mesenchyme expression persisted until PAFF regression (around stages 45 and 46). From stage 39, we observed *hand2* expression in the core of the nascent pelvic fins (Extended Data Fig. 7k,l). We conclude that a larval PAFF with an LPM mesenchyme is ancestral for actinopterygians. To establish if an LPM contribution to median fins occurred before the origin of paired fins, we examined the larvae of a jawless vertebrate, the sea lamprey (*Petromyzon marinus*).

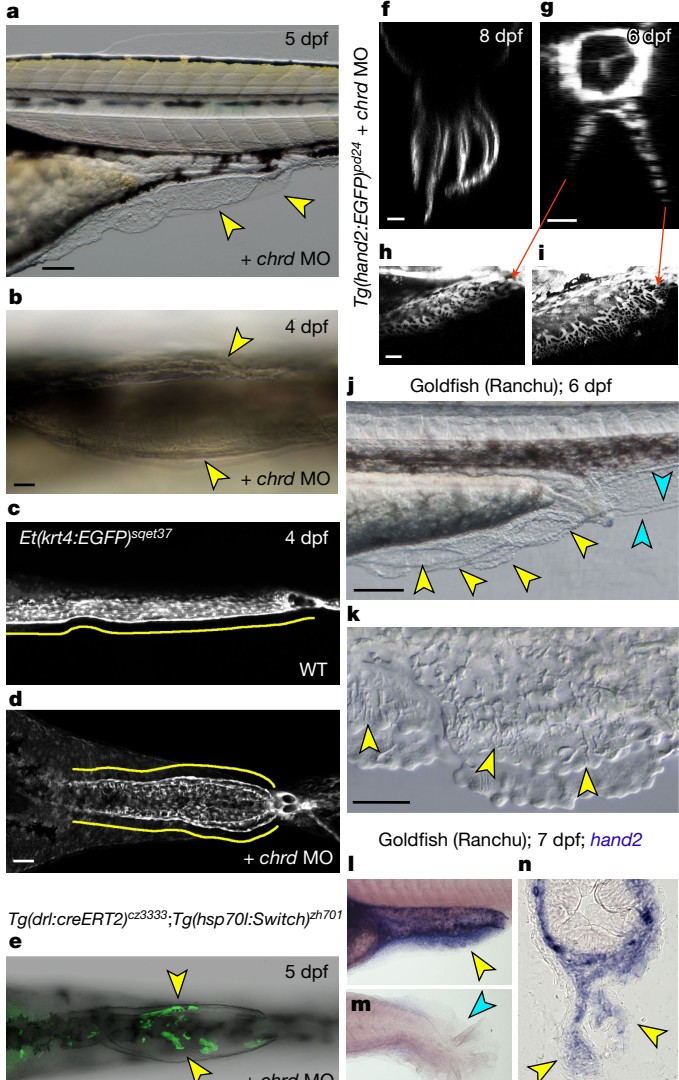

**Fig. 4 | Duplication of the PAFF into paired fin folds. a**,**b**, Lateral (**a**) and ventral (**b**) Nomarski images of PAFFs of 5 dpf (**a**) and 4 dpf (**b**) embryos injected with MO targeting *chordin*, resulting in duplication of PAFF (yellow arrowheads). **c**,**d**, Confocal micrographs, imaged ventrally, of PAFFs in 4 dpf *ET37* transgenic larvae uninjected (**c**) or injected with a *chrd* MO (**d**). PAFFs are indicated with yellow lines. **d**, Note the duplicated anal openings following *chrd* MO injection. **e**, Ventral confocal image of *chrd* MO duplicated PAFFs of the transgenic line shown above. Hydroxytamoxifen treatment at the 12-somite stage and heat shock before imaging show that mesenchyme of duplicated PAFFs is derived from the LPM (yellow arrowheads). **f**–**i**, Light-sheet (**f**) and confocal (**g**–**i**) images of *Tg(hand2:EGFP)* larvae at 8 dpf (**f**) and 6 dpf (**g**–**i**). Orthogonal display through the *x*–*z* plane (**f**,**g**) shows that multiple PAFFs can form and that duplicated PAFFs contain eGFP-positive mesenchyme (**g**–**i**). Lateral views of left (**h**) and right (**i**) duplicated PAFFs of sample in **g** are given. **j**,**k**, Lateral low- (**j**) and high-power (**k**) Nomarski images of the PAFF of the Ranchu goldfish at 6 dpf. The multiple PAFFs (**j**) and individual mesenchyme cells (**k**) are indicated by yellow arrowheads. Duplicated caudal fin folds are indicated by cyan arrowheads (**j**). **l**–**n**, Lateral (**l**,**m**) and transverse (**n**) views of pre-anal (**l**,**n**) and caudal (**m**) fin folds of 7 dpf Ranchu larvae stained by in situ hybridization for *hand2*, where *hand2*-positive PAFF mesenchyme is indicated by yellow arrowheads. Absence of *hand2* in the caudal fin fold is indicated by the cyan arrowhead (**m**). Scale bars, 100 µm (**a**,**e**,**m**); 20 µm (**b**,**f**–**h**,**n**); 50 µm (**d**,**k**); 200 µm (**j**).

Lampreys also possess a small transient larval PAFF[25], within which strong, specific expression of the lamprey *hand2* orthologue, *HandA*, was detected (Fig. 3f,g), although there was evidence of fainter *HandA*

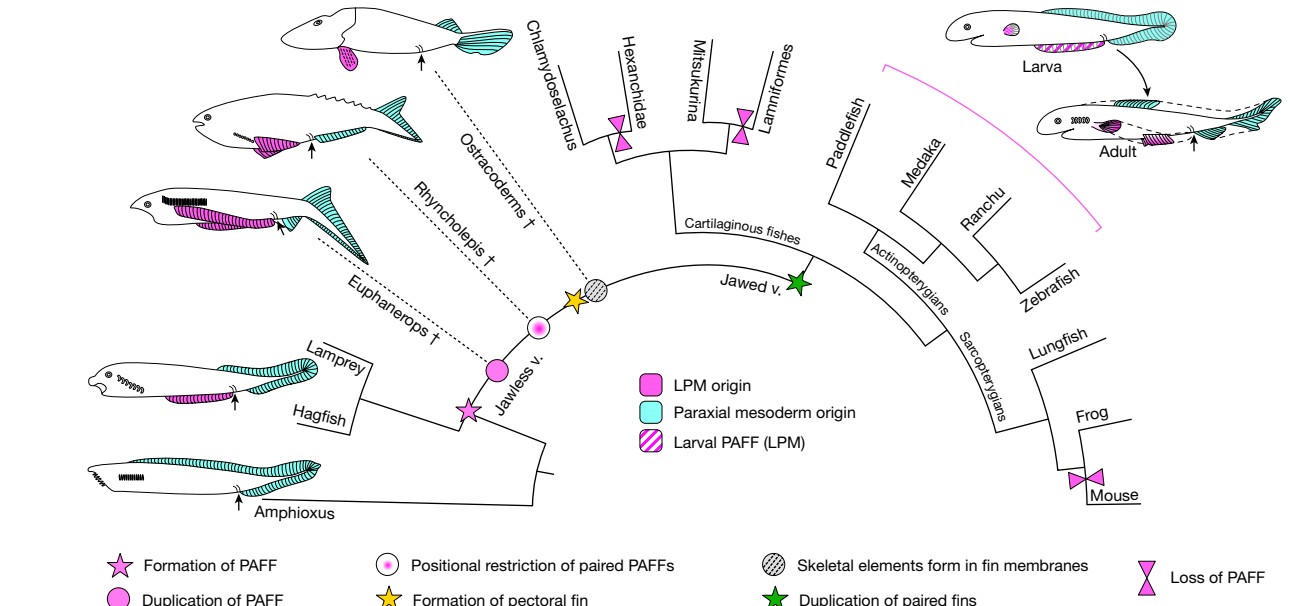

**Fig. 5 | Hypothesis of the elaboration of the PAFF to paired fins.** Simplified evolutionary scenario of vertebrates showing the presence of a PAFF and subsequent modifications leading to paired fins. Dashed lines and dagger symbols indicate extinct lineages, and solid lines indicate extant lineages. PM-derived fins and fin folds are in cyan, while LPM-derived fins are in pink. Larval PAFF is hatched. Black arrows indicate the position of the anus.

Legend:
- ★ (pink star) Formation of PAFF
- ● (pink circle) Duplication of PAFF
- ○ (light pink circle) Positional restriction of paired PAFFs
- ★ (yellow star) Formation of pectoral fin
- ◍ (hatched circle) Skeletal elements form in fin membranes
- ★ (green star) Duplication of paired fins
- ◁▷ (pink bowtie) Loss of PAFF
- ▦ LPM origin
- ▦ Paraxial mesoderm origin
- ▨ Larval PAFF (LPM)

expression in the dorsal anterior fin. Finally, both the embryos of the sarcopterygian lungfish[26] and amphibian tadpoles (e.g., *Xenopus laevis*[27], *Xenopus tropicalis* and axolotls[15]) also possess a small transient PAFF (Fig. 3h). Fluorescence in situ hybridization demonstrated individual *hand2*-positive mesenchymal cells invading this fin in *X. tropicalis* but not the major median fin lobe at stage 42 (Fig. 3h–j). We confirmed this LPM contribution by injecting *-6.35drl:EGFP* into *X. laevis* embryos. *X. laevis* has a larger PAFF than *X. tropicalis,* and we observed extensive transient eGFP labelling in the PAFF compared with the caudal fin fold (Extended Data Fig. 7m–o). Thus, we show an LPM origin for PAFF mesenchyme in the larvae of representatives of major extant vertebrate linages: cyclostomes, actinopterygians and sarcopterygians.

Given the LPM origin of the median PAFF, we considered that it may have importance in the transition from PM-derived median fins to LPM-derived paired fins. We tested if this LPM median fin fold could be duplicated into paired LPM-derived lateral fin folds. Zebrafish larvae mutant for the bone morphogenetic protein (BMP) antagonist *chordin* develop ventral caudal fin fold duplications, although whether the PAFF is duplicated in these mutants has not been reported[28]. Injection of a low dose of *chrd* MO into zebrafish recapitulated various ventralized phenotypes, which included individuals with duplicated or multiple PAFFs (Fig. 4a,b,f and Extended Data Fig. 8a,b). We visualized these PAFF phenotypes by *chrd* MO injection into the *ET37* transgenic line, which demonstrated that each fin fold of duplicated PAFFs contained mesenchyme (Fig. 4c,d). Lineage tracing using the *Tg(drl:creERT2; hsp70l:Switch)* transgenic combination injected with *chrd* MO demonstrated that the mesenchyme cells of the duplicated PAFFs are indeed derived from the LPM (Fig. 4e). By generating *chrd* morphants in the *Tg(hand2:EGFP)* line and then imaging by light-sheet microscopy, we generated three-dimensional reconstructions documenting duplicated PAFFs along the yolk extension, all of which harboured eGFP-positive mesenchymal cells, further demonstrating that the mesenchyme of these duplicate fin folds was LPM derived (Fig. 4g–i, Extended Data Fig. 8c–j and Supplementary Video 1). Of note is the generation of multiple parallel fin folds in individual embryos following *chrd* MO injection (Fig. 4f). These observations indicate that bifurcation of the LPM-seeded PAFF readily arises from modulating ventral BMP signalling, such as by reducing the dose of the dorsal inhibitor Chordin.

To determine if the generation of multiple PAFFs could have evolved spontaneously as part of natural variation, we examined the twin-tail goldfish strain, Ranchu, which has been shown to display bifurcated fin folds including PAFFs due to a loss-of-function mutation in a *chordin* paralogue (*chdA*$^{E127X}$)[29–31]. The bifurcated PAFFs and caudal fin folds of larval Ranchu appeared similar to those of zebrafish larvae injected with low doses of *chrd* MO (Fig. 4j and Extended Data Fig. 8k,l). These PAFFs were all populated by mesenchymal cells (Fig. 4k). Using *hand2* in situ hybridization as a proxy of LPM lineage to infer the origin of the Ranchu PAFF mesenchyme, we observed that the core of both duplicated PAFFs expressed *hand2*, whilst the caudal fin fold did not (Fig. 4l–n and Extended Data Fig. 8m,n). This suggests that, as in the zebrafish PAFF, the bifurcated Ranchu PAFFs are also LPM derived. Notably, in addition to those displaying simple PAFF duplication, some individuals had three or more parallel PAFFs (Fig. 4j and Extended Data Fig. 8n). Thus, we conclude that LPM-derived paired fin folds that can be readily generated in zebrafish larvae have also arisen spontaneously in twin-tail goldfish; thus, they represent a viable morphological innovation. Notably, duplicated LPM-derived PAFFs were situated in ventrolateral locations between the future pectoral and pelvic fin domains, overlying the proposed competence stripes of appendage formation in gnathostomes[32].

Combining several models and techniques, we here have identified a larval median fin with a mesenchyme core derived from LPM and propose it as an intermediate between PM-derived median fins and LPM-derived paired fins. While apparently conserved from cyclostomes to amphibia, most extant chondrichthyan embryos do not appear to have a discernible PAFF. We used micro-computed tomography (microCT) to examine prehatching stages of the elasmobranch Epaulette shark *Hemiscyllium ocellatum*, which did not show the presence of a clear pre-anal fin (Extended Data Fig. 9a–i). Although this pattern appears to support a loss of the PAFF somewhere in the chondrichthyan lineage, we cannot rule out the possibility that a PAFF arose independently in cyclostomes and Osteichthyes. It is interesting to note that embryos and adults of the frilled shark, *Chlamydoselachus*, do show bifurcated tropeic folds in the ventral midline (Extended Data

Fig. 9j–m), structures previously invoked as supporting the fin fold hypothesis[33].

PAFFs are mostly transient larval structures that generally lack a mineralized skeleton, which may explain their poor documentation in the fossil record. Although a true PAFF does not persist into adulthood in most species, it is seen in hagfish adults, where unlike the caudal median fin, it does not possess cartilage rays[34]. In the fossil record, a pre-anal fin is seen in some stem vertebrate fossils, including *Haikouella* and *Haikouichthys*, while the pre-anal fin of *Kerreralepis* shows well-developed plates[35,36]. Zebrafish larval fin mesenchyme is known to persist and contribute to the lepidotrichia of adult fins[18], and LPM-derived dermal fibroblasts form cartilage during axolotl limb regeneration[37]. These data suggest the potential of the PAFF mesenchyme to generate skeletal outcomes.

Spontaneous duplication in twin-tail goldfish species indicates that paired LPM-derived PAFFs could have readily arisen during evolution. In Ranchu, we frequently observed multiple PAFFs, permitting retention of a median PAFF and divergence of duplicates into paired fins. Although there is no evidence of lateral fin folds present in extant vertebrates, adult specimens of the hagfish species *Neomyxine biniplicata* possess anterior lateral paired folds that terminate close to the pre-anal fin[38], resembling partial duplications of the pre-anal fin, although their relevance to paired fin evolution is disputed[39]. In the fossil record, there is evidence that both mineralized and unmineralized lateral paired fin folds were common in anaspids, although homology of anaspid and gnathostome paired fins is highly contentious, as is their relevance to paired fin evolution and the lateral fin fold hypothesis[1,11]. *Pharyngolepis*, *Cowielepis* and *Euphanerops* display pre-anal ventrolateral paired ribbon structures or triangular fins with radials, while in *Jamoytius*, the ribbons lacked mineralized structures[12,40,41]. The galeaspid, *Tujiaaspis*, has been recently described bearing skeletal ventrolateral fins composed of skeletal units[42]. Subsequent regionalization of these adult paired elongate fins through anterior restriction to the pectoral fin region is proposed and exemplified by *Pharyngolepis* and *Rhyncholepis*[11,43].

Our work gives weight to a model for paired fin evolution through co-option of a larval median fin programme to the LPM followed by fin duplication and subsequent regionalization to pectoral and pelvic fins. We present one possible model for how the PAFF may have led to the generation of LPM-derived paired fins (Fig. 5). The PAFF may have originated as a small ectoderm-only fin fold (as seen in amphioxus larvae[44]) with an LPM contribution evolving subsequently upon alterations in LPM topology, such as persistence of a somatopleure and/or a lateral mesodermal divide[45,46]. Similar LPM tissue contexts may have then led to PAFF elongation and following duplication, paired fin regionalization. As the PAFF possesses characteristics of both unpaired and paired fins, the PAFF may represent a novel evolutionary module or at least demonstrate components of the developmental mechanisms that contributed to the emergence of paired appendages.

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

## Methods

### Animal husbandry and experiments

Adult zebrafish and medaka were maintained under standard conditions, and embryos were obtained through natural crosses, kept in E3 medium at 28.5 °C and staged according to Iwamatsu[47] and Kimmel et al.[48]. Embryos of twin-tail goldfish (Ranchu strain) were purchased from a local breeder in Singapore, cultured in E3 medium (with methylene blue) at room temperature and staged as per Li et al.[49]. The experimental procedures and protocols were approved by the institutional animal care and usage committees (IACUCs) at the Institute of Molecular and Cell Biology, Agency for Science, Technology and Research, Singapore (IACUC number 140924); Nanyang Technological University (IACUC number A18002); University of Zurich and the veterinary office of the Canton of Zurich; and the University of Colorado School of Medicine Anschutz Medical Campus (protocol number 979). Paddlefish embryos (*P. spathula*) were obtained and staged as previously described[8], and experiment protocols and animal care were approved by James Madison University (IACUC number 20-1601). Lamprey embryo work was performed based on published protocols and staged per Tahara[50], with experimental protocols approved by the IACUC of the California Institute of Technology (IACUC number 1436). Experiments involving *X. tropicalis* and *X. laevis* were approved by the local ethics committee at the University of Manchester and the Home Office (number PFDA14F2D). *X. laevis* embryos were injected with 100 pg of the *-6.35drl:EGFP* construct[24] together with 400 pg of in vitro-transcribed membrane−*mCherry* messenger RNA in each blastomere at the two-cell stage. Medaka embryos were obtained from the National University of Singapore under IACUC numbers BR19-0120 and BR22-1497. Embryos of Epaulette shark were obtained from brood stock at Monash University as approved by the Monash Animal Ethics Committee under license (number 30347) and staged per Ballard et al.[51].

### Fish lines and MOs

The zebrafish WT strain used in this study was AB, while the following mutants were used: *frf*[tp34] (ref. 20) and *han*[S6] (ref. 22). The transgenic lines used were *ET37*, *Tg(lyve1b:DsRed2)*[nz101], *Tg(tbx16l:GAL4-VP16)*[sq13]; *Tg(UAS:Kaede)*[rk8], *Tg(hand2:EGFP)*[pd24] and *Tg(drl:creERT2)*[cz3333]; *Tg(hsp70l:switch)*[zh701] as described previously[18,23,24,52]. The *drl:H2B-Dendra2* transgenic line was generated by *Tol2*-mediated transgenesis using a plasmid construct assembled through the Multisite Gateway system with LR Clonase II Plus (Life Technologies) and components from the Tol2kit[53]. The medaka used was the Cab strain, and the *-6.35drl:EGFP* transgenic line was generated by *I-SceI* meganuclease-mediated transgenesis[54] using a plasmid that contained the *-6.35drl:EGFP* sequence[24] flanked by two I-SceI recognition sites. The *hand2* MO sequence was (5′–3′) CCTCCAACTAAACTCATGGCGACAG and injected at 500 µM. The *chordin* MO sequence was (5′–3′) ATCCACAGCAGCCCCTCCATCATCC and injected at 200 µM.

### In situ hybridization

In situ hybridization on zebrafish, medaka and goldfish embryos was conducted as described[55]. Probes used were *fbln1* (ref. 56), *itgb3b*[57], *bmp1a*[20], zebrafish *hand2* (ref. 22) and medaka *hand2* (ref. 58). Goldfish *hand2* probe was generated by polymerase chain reaction using goldfish complementary DNA and the primer sequences (5′–3′) ACGTTTTATGGGGAGACAACC and <u>TAATACGACTCACTATAGGG</u>TCTT CCTTGGCGTCTGTCTT, where the T7 RNA polymerase site is underlined. The transverse section of whole-mount in situ hybridization was achieved via cryosections. Briefly, stained embryos were embedded in agar sucrose and cryoprotected in 30% sucrose. Samples were embedded in OCT medium (Tissue-Tek) and cut into 16- to 20-µm sections in a Leica cryostat (CM3050S). Paddlefish in situ hybridization was performed as previously stated, and the paddlefish *hand2* probe was used[59]. Whole-mount chromogenic in situ hybridizations and fluorescent in situ hybridizations and sections were performed as previously described[60,61] on *X. tropicalis* larvae using a *hand2* antisense probe from the expressed sequence tag (EST) clone (TTba012k13) identified from the *X. tropicalis* full-length EST library[62]. Lamprey in situ hybridization was accomplished using *HandA* as a probe according to the published protocol[63].

### Transplantation

The *hand2* cell autonomy experiments were performed by transplanting marginal cells at gastrula stages from *ET37* hosts to the margin of unlabelled WT hosts as previously described[64]. Donors were injected with 50 pg *H2B−mCherry* messenger RNA either with or without 500 µM *hand2* MO. Chimeras were then raised, and PAFF mesenchyme was imaged at 3 dpf.

### DiI injection of medaka

CellTracker CM-DiI Dye (C7000; Invitrogen) was microinjected using filamented glass capillaries at a concentration of 1 µg µl⁻¹ into the LPM, just posterior to Kupffer's vesicle at the approximately four-somites stage using Eppendorf FemtoJet.

### Immunostaining

Antibody staining was executed as previously described[20] using either zebrafish embryos or sections. The primary antibodies, sources and dilutions used were anti-Col2A1 (II-II6B3; DSHB; 1:100), anti-eGFP (TP401; Torrey Pines; 1:500), zns-5 (AB_10013796; ZIRC; 1:200) and anti-SM22 alpha/Transgelin (ab14106; Abcam; 1:250). Secondary antibodies were all from Invitrogen and used at 1:500: Alexa 488-conjugated donkey anti-rabbit immunoglobulin G (IgG; catalogue number A-21206), Alexa 546-conjugated donkey anti-mouse IgG (catalogue number A10036), Alexa 647-conjugated donkey anti-rabbit IgG (catalogue number A-31573) and Alexa 647-conjugated donkey anti-mouse IgG (catalogue number A-31571).

### Microscopy

Confocal images were taken on an LSM800, LSM880 or LSM980 Zeiss microscope. For high magnification of bright-field or Nomarski images, a Zeiss AxioImager M2 was used. A Zeiss AxioZoom V16 was utilized to capture low magnification of bright-field images. Light-sheet fluorescence microscopy was undertaken using a Zeiss Lightsheet Z.1, and sample preparation and experimental procedures were performed based on the manufacturer's manual. Microscopy images (fluorescence and bright field) were analysed by ZEN Blue v.3.6 (Zeiss), Fiji (ImageJ v.1.52p) and IMARIS v.9.9.1 (Oxford Instruments).

### High-resolution X-ray computed tomography

Shark samples previously fixed in 4% paraformaldehyde (PFA) and dehydrated for freezer storage were stained with 1% $I_2$ dissolved in ethanol according to Metscher[65]. X-ray computed tomography scans were done using a Zeiss Xradia 520 Versa system. Images were generated using Avizo 3D software (ThermoFisher Scientific).

### Photoconversion

To track cell lineage, Kaede and Dendra-2 photoconversion was conducted as previously described[18]. In brief, 24 hpf Kaede or Dendra-2 green-expressing embryos were mounted in 3% methyl cellulose or 0.5% low-melting point agarose and imaged by confocal microscopy. A selected region of interest was photoconverted to express red fluorescence using 405-nm ultra-violet laser illumination. Embryos were imaged immediately after photoconversion by sequential acquisition using 488- and 561-nm-wavelength channels to confirm successful conversion. Then, the same embryos were re-imaged at 40, 48 or 72 hpf.

## Tamoxifen treatment

To induce Cre activity in *Tg*(*drl:creERT2, hsp70l:Switch*) transgenic embryos, 4-OHT was added in E3 (final concentration of 10 μM) at the 12-somite stage. Then, 4-OHT was washed off at 24 hpf, and embryos were raised until the desired stage. For specific embryonic and larval time points, embryos were heat shocked for 1 h at 37 °C to initiate the expression of eGFP (2–3 h before fixation). Subsequently, embryos were sampled, fixed and stained with 4′,6-diamidino-2-phenylindole (DAPI), and confocal imaging was undertaken.

Adult zebrafish were heat shocked for 16 h at 37 °C, anesthetized with Tricaine and euthanized in an ice-water bath. Successfully switched individuals were selected based on eGFP-expressing heart (major derivate organ of the *drl*-expressing LPM), and the paired pectoral and pelvic fins and anal fin were dissected and fixed overnight with 4% PFA.

## Statistical analysis and reproducibility

Statistical analysis was performed using Prism v.9 (GraphPad). A two-tailed Mann–Whitney test was used when two conditions were compared. No statistical methods were used to predetermine sample size. All experiments were performed at least three times on different weeks with different biological samples. The experiments were not randomized, and investigators were not blinded. Consistent labelling within each batch of embryos was confirmed, and representative samples were used for imaging.

## Reporting summary

Further information on research design is available in the Nature Portfolio Reporting Summary linked to this article.

## Data Availability

The authors confirm that all relevant data are provided in this paper and in its Extended Data files. The data for measurements of fin size and cell number in Extended Data Fig. 3m–o are available in Figshare with the identifier https://doi.org/10.6084/m9.figshare.22269769. Source data are provided with this paper.

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

**Acknowledgements** We thank K. Poss and M. Tanaka for providing zebrafish and medaka *hand2* plasmids, respectively; Z. Gong for the transgenic fish line *TgBAC(hand2:EGFP)*; D. Traver for input on the first iteration of Dendra2 transgenics; Y.-K. Tea for input; and K. P. Lim of the Lee Kong Chian Natural History Museum, Singapore for assisting with the frilled shark specimen. We also thank the Nanyang Technological University Zebrafish facility, the NTU Optical Bio-Imaging Centre (NOBIC), Colorado University (CU) Anschutz zebrafish care staff and all past and present members of our teams for continued support. This work was funded by the Industry Aligned Fund (IAF) Agency for Science, Technology and Research (grant to T.J.C. and K.-W.T.); Ministry of Education (MoE) Tier 3 (grant 2016-T3-1-005 to T.J.C., C.W. and H.M.); Ministry of Education (MoE) Tier 1 (grant 2016-T1-001-055 to T.J.C. and C.Z.); Ministry of Education (MoE) Tier 2 (grant MOE-T2EP30221-0008 to C.W.); the Company of Biologists (travelling fellowship to M.J.T.); the National Science Foundation (grants IOS-1853949 to M.C.D. and 2203311 to C.M.); the Swiss National Science Foundation Sinergia (grant CRSII5_180345 to C.M.); the Swiss Bridge Foundation (C.M.); Additional Ventures Single Ventricle Research Fund (SVRF) (grant 1048003 to C.M.); the University of Colorado School of Medicine Anschutz Medical Campus and the Children's Hospital Colorado Foundation (C.M.); the National Institutes of Health (NIH), National Institute of General Medical Sciences (grants 1T32GM141742-01 to H.R.M. and 3T32GM121742-02S1 to H.R.M.); Australian Research Council (discovery grant DP200103219 to F.J.T. and P.D.C.); National Health and Medical Research Council (senior principal research fellow APP1136567 to P.D.C.); and the NIH (grant R35NS111564 to J.S. and M.E.B.).

**Author contributions** Experiments were conceived and designed by T.J.C., C.M., K.-W.T., M.E.B., M.C.D., E.A., P.D.C., C.W. and H.H.R. and were performed and analysed by K.-W.T., R.L.L., K.D.P., J.S., H.M., H.R.M., A.N.C., Y.L., R.L., K.D., M.J.T., C.Z., E.C.B., W.T.W. and F.J.T. The manuscript was written by K.-W.T. and T.J.C., with editing and input from C.M., J.S., A.N.C., M.C.D., F.J.T., P.D.C., R.L.L., K.D.P. and E.A.

**Competing interests** The authors declare no competing interests.

**Additional information**
**Correspondence and requests for materials** should be addressed to Christian Mosimann or Tom J. Carney.

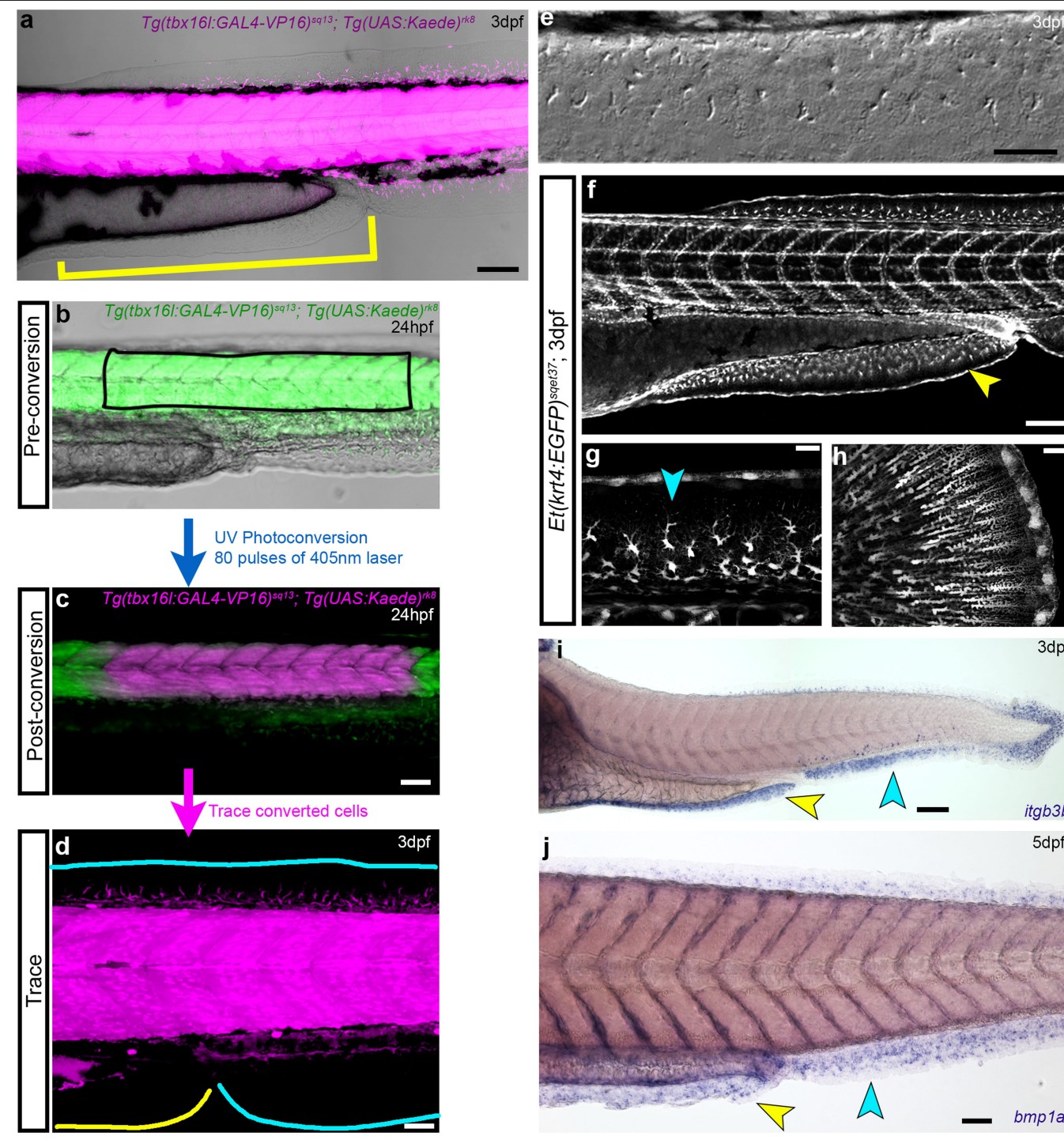

**Extended Data Fig. 1 | A non-paraxial mesoderm derived median fin fold.**
**a**, Confocal image of 3 dpf *Tg(tbx16l:GAL4-VP16); Tg(UAS:Kaede)* embryo with paraxial mesoderm labelled by photoconverted Kaede (magenta). Unlabelled PAFF underscored by yellow bracket. **c-d**, Expression of Kaede in the dorsal caudal fin fold of 3 dpf *Tg(tbx16l:GAL4-VP16); Tg(UAS:Kaede)* is not due to localised de novo expression from the *tbx16l* promoter. Confocal images of 24 hpf trunks of both prior to (**b**) and after (**c**) UV photoconversion. Unconverted Kaede is in the green channel overlaid with converted Kaede in magenta. Region of paraxial mesoderm conversion and Nomarski image given

in (**b**). At 3 dpf, converted cells can be seen in the adjacent fin fold dorsally indicating Kaede is reporting lineage (**d**). **e**, Nomarski image of PAFF at 3 dpf showing presence of mesenchymal cells. **f**–**h**, Confocal images of the pre-anal fin fold (**f**), dorsal (**g**) and caudal (**h**) regions of the caudal fin fold of the *ET37* Enhancer Trap transgenic line at 3 dpf, indicating PAFF contains numerous mesenchymal cells which are morphologically comparable to mesenchyme of other larval fin folds. **i**–**j**, In situ hybridisation of fin mesenchyme markers *itgb3b* (**i**; 3 dpf) and *bmp1a* (**j**; 5 dpf) in both PAFFs (yellow) and caudal fin folds (cyan arrowhead). Scale Bars: 100 μm (**a**,**f**,**i**), 50 μm (**c**,**d**,**e**,**j**), 20 μm (**g**,**h**).

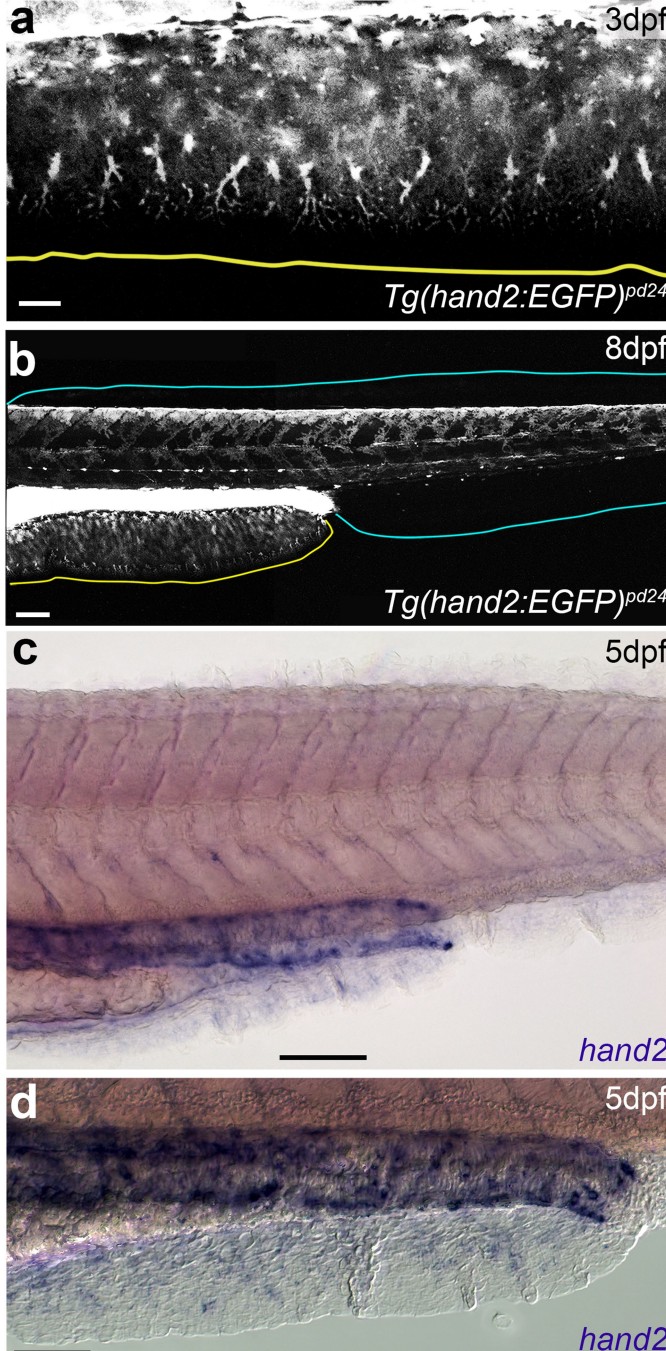

**Extended Data Fig. 2 | Expression of *hand2* in the pre-anal fin fold.**
**a**–**b**, High (**a**) and low (**b**) magnification confocal images of *Tg(hand2:EGFP)*
zebrafish embryos at 3 dpf (**a**) and 8 dpf (**b**) showing eGFP labelling of
mesenchyme of the PAFF (yellow outline, **a**,**b**), but not the caudal fin fold (cyan
outline, **b**). **c**–**d**, In situ hybridisation of *hand2* at 5 dpf showing expression of
*hand2* only in the PAFF (**c**). Higher magnification with Nomarski optics indicates
expression in mesenchyme (**d**). Scale Bars: 100 µm (**b**,**c**), 50 µm (**d**), 20 µm (**a**).

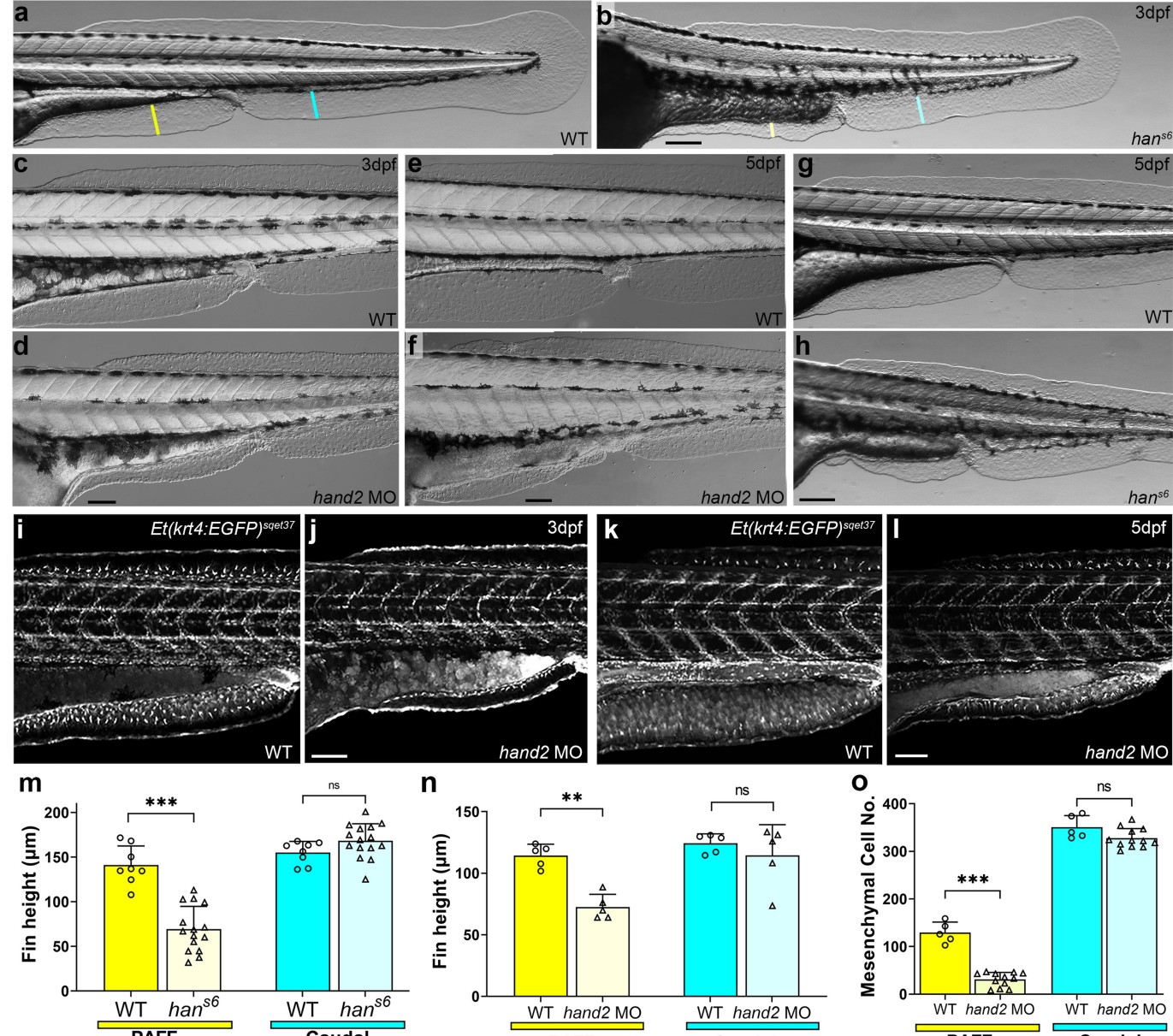

**Extended Data Fig. 3 | Loss of Hand2 leads to reduction of pre-anal fin folds but not caudal fin folds. a–l,** Lateral Nomarski images of uninjected WT (**a,c,e,g**), *han^s6* mutants (**b,h**) and *hand2* morphants (**d,f**) at 3 dpf (**a–d**) or 5 dpf (**e–h**). **i–l,** Confocal images of 3 dpf (**i,j**) and 5 dpf (**k,l**) *ET37* Enhancer Trap transgenic larvae, uninjected (**i,k**) or injected with a *hand2* morpholino (**j,l**). **m,n** Quantification of 3 dpf fin height of PAFFs (yellow) and caudal fin folds (cyan) in uninjected WT, *hand2* mutants (*han^s6*; **m**) and *hand2* morphants (**n**). Loss of Hand2 leads to significant reduction of PAFFs but not caudal fin folds. n = 8 (WT), n = 15 (*han^s6*), n = 5 (*hand2* MO). Location of measurements of PAFF and caudal fin shown in (**a**) and (**b**) by yellow and cyan lines respectively.

***: p = 0.000008; **: p = 0.008; ns: p = 0.12 (**m**); ns: p > 0.9999 (**n**). **o,** Quantification of mesenchymal cell numbers in pre-anal (yellow bars) and ventral caudal (cyan bars) fin folds at 3 dpf. There is a significant reduction of PAFF mesenchyme in *hand2* morphants compared to WT, but ventral caudal fin mesenchymal cell numbers are unaffected. n = 5 (WT), n = 12 (*hand2* MO). ***: p = 0.000323; ns: p = 0.06. **m–o:** Data are presented as mean values with error bars representing standard deviations. In all cases n refers to biologically independent embryos. Two-sided Mann-Whitney test. Scale Bars: 100 μm (**d,f,j,l**), 200 μm (**b,h**).

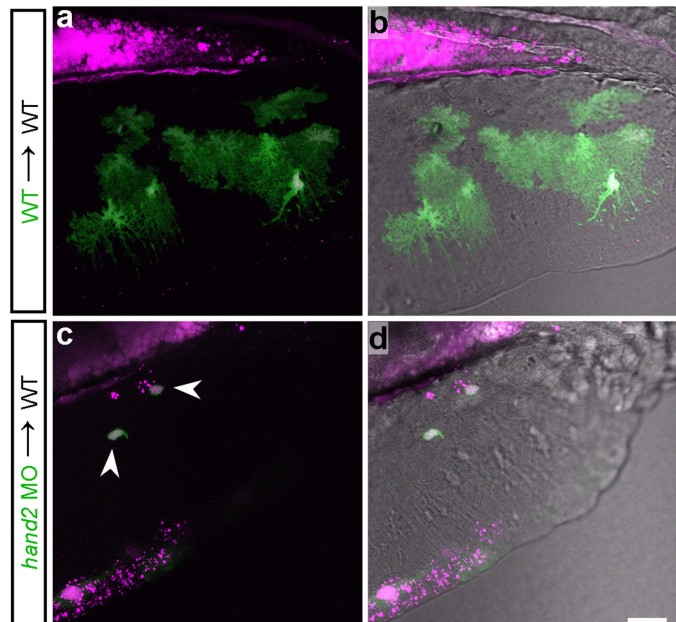

**Extended Data Fig. 4 | Hand2 acts cell autonomously in PAFF mesenchyme.**
**a**–**d**, Lateral confocal images of PAFF of 3 dpf embryos. Cells were transplanted
into WT hosts from *ET37* embryos that were injected with *H2B-mCherry* mRNA
(**a**,**b**) or co-injected with *H2B-mCherry* mRNA and 500 µM *hand2* MO (**c**,**d**).
Fluorescent confocal images of mCherry (magenta) and eGFP (green) (**a**,**c**) are
overlayed on Nomarski image (**b**,**d**). Arrowheads indicate transplanted cells
from *hand2* MO injected donors (**c**). Scale Bar: 20 µm (**d**).

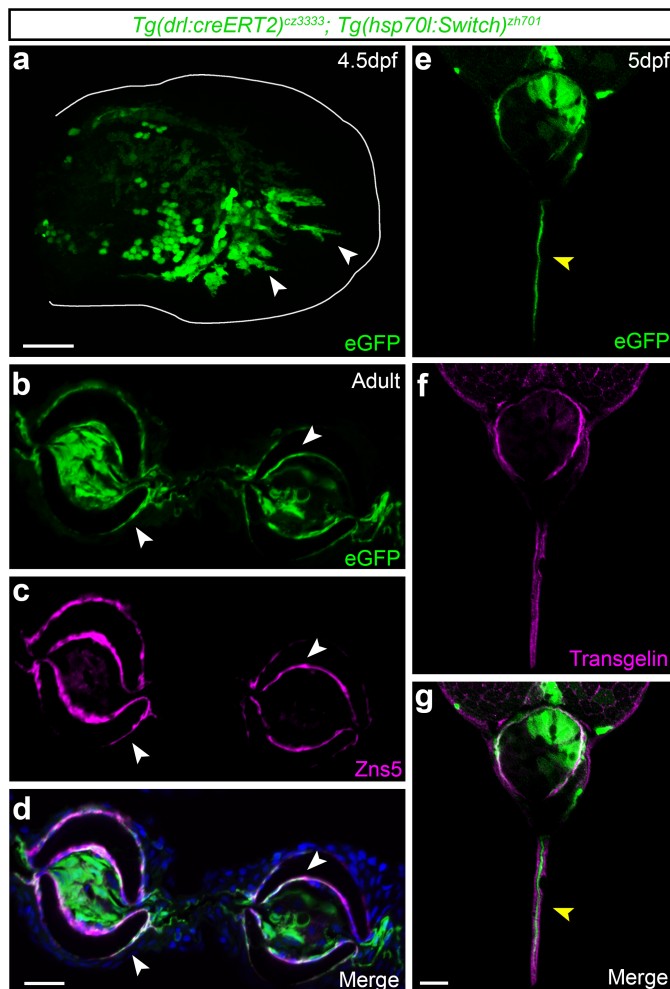

**Extended Data Fig. 5 | Lineage tracing of the LPM into paired fins and the PAFF. a**–**g**, Permanently labelled LPM cells can be seen within pectoral and pelvic fins and pre-anal fin fold (arrowheads). Confocal images of the pectoral fin at 4.5 dpf (**a**), adult pelvic fin (**b**–**d**) and pre-anal fin fold at 5 dpf (**e**–**g**) of *Tg(drl:creERT2); Tg(hsp70l:Switch)* transgenics following Hydroxytamoxifen treatment and heat-shock prior to imaging. Pelvic fins and PAFF are shown as transverse sections (**b**–**g**) and were fluorescently immunostained for eGFP (**b**,**e**), Zns5 (**c**) or Transgelin (**f**). Merged images are shown in (**d**,**g**). LPM-derived mesenchyme cells of the pectoral and pre-anal fin folds are indicated by white and yellow arrowheads respectively (**a**, **e**–**g**). LPM-derived Zns5-positive osteoblasts of the pelvic fins are indicated by white arrowheads (**b**–**d**). Scale Bars: 20 μm (**d**,**g**), 50 μm (**a**).

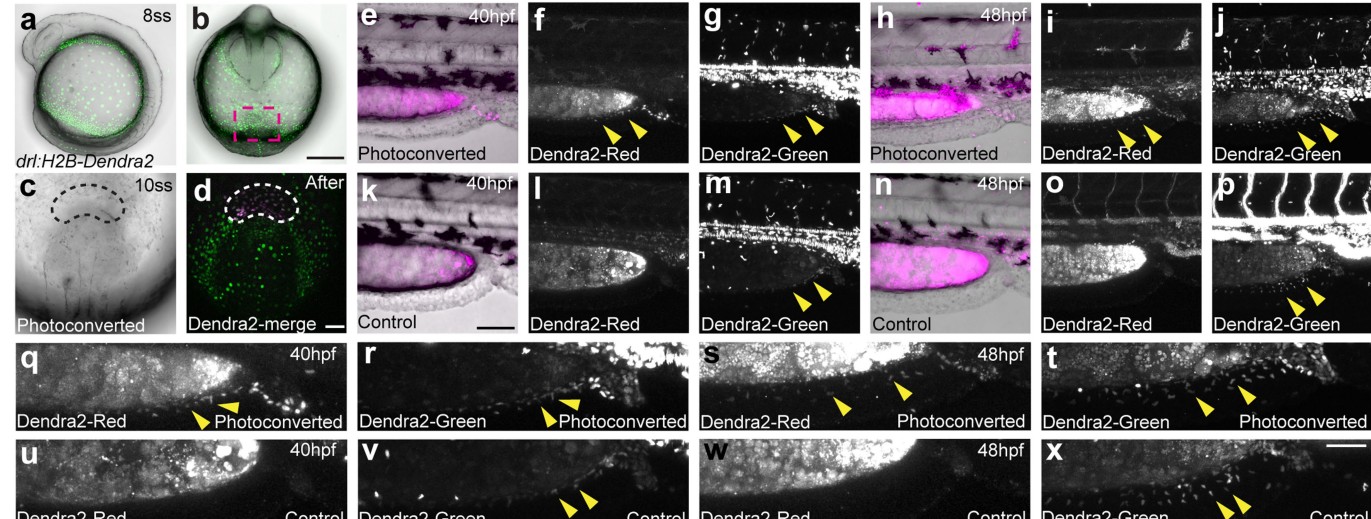

**Extended Data Fig. 6 | Pre-anal fin fold mesenchyme originates from posterior-most *drl*-positive cells. a–d**, Lateral (**a**) and ventral (**b**–**d**) views of 8ss (**a**–**b**) or 10ss (**c**–**d**) *drl:H2B-Dendra2* transgenic line either prior to (**a**–**b**) or following (**c**–**d**) photoconversion of posterior-most LPM (region outlined by dotted line) using UV laser illumination. **e**–**p**, Lateral confocal images of the PAFF region at 40 hpf (**e**–**g**, **k**–**m**) and 48 hpf (**h**–**j**, **n**–**p**) of a photoconverted embryo (**e**–**j**) and an unconverted control (**k**–**p**). Panels show Dendra2-red (**f,i,l,o**), Dendra2-green (**g,j,m,p**) and merged (**e,h,k,n**) channels. PAFFs are indicated by yellow arrowheads, and are visible in green channel in both converted and controls, but only converted embryos show red labelled PAFFs (**f,i**; n = 4 photoconverted fish). Magnified views of panels (**f,g,i,j,l,m,o,p**) are given in panels (**q**–**x**) respectively. Scale Bars: 200 μm (**b**), 50 μm (**d,x**), 100 μm (**k**).

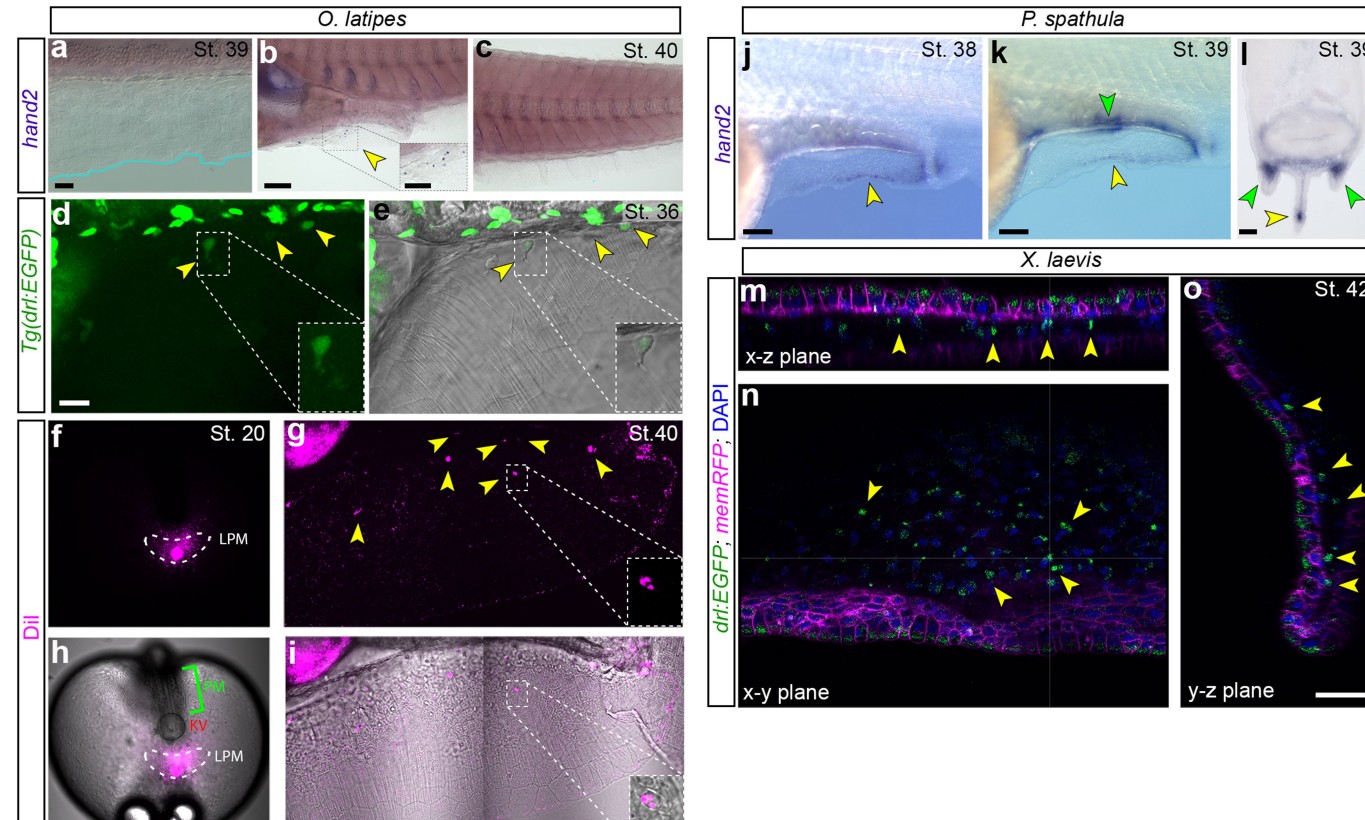

**Extended Data Fig. 7 | LPM origin of PAFF mesenchyme is conserved in medaka, paddlefish and *Xenopus*. a–c**, In situ hybridisation of medaka *hand2* at Stage 39 (**a**) and Stage 40 (**b–c**) showing exclusive expression in pre-anal fin mesenchyme (yellow arrowhead, **b**), but not in the caudal fin mesenchyme (cyan outline). Ventral caudal fin imaged in (**a**). **d–e**, Lateral confocal images of PAFF of Stage 36 transgenic *Tg(-6.35drl:EGFP)* medaka embryo. Fluorescent confocal images of eGFP (**d**) are overlayed on Nomarski image (**e**). Faint eGFP perdurance is seen in nascent PAFF mesenchymal cells (yellow arrowheads and inset). **f–i**, Ventral (**f**,**h**) and lateral (**g**,**i**) confocal images of medaka embryo injected with DiI in the posterior LPM at Stage 20 (**f**,**h**) and traced to PAFF at Stage 40 (**g**,**i**). DiI within the mesenchyme indicated by yellow arrowheads, with higher magnification example shown in inset (**g**,**i**). DiI signal (**f**,**g**) is overlayed with Nomarski (**h**,**i**). Location of paraxial mesoderm (PM), Kupffer's vesicle (KV) and region of posterior LPM are indicated in green, red, and white respectively in (**h**). **j–l**, In situ hybridisation of *hand2* in Stage 38 (**j**) and Stage 39 (**k–l**) paddlefish embryos imaged laterally (**j**,**k**) or in transverse section (**l**). *hand2*-positive PAFF fin mesenchyme indicated by yellow arrowheads is located distally. Expression of *hand2* is also seen in the nascent pelvic fins at Stage 39 (green arrowheads **k**,**l**). **m–o**, Confocal images of a region of *Xenopus laevis* PAFF at NF Stage 42 injected with -*6.35drl:EGFP*. Lateral x-y view (**n**) is shown with orthogonal sections in the x-z plane (**m**) and y-z plane (**o**). Transient mosaic expression in the interstitial mesenchyme is highlighted with yellow arrowheads (**m–n**). Scale Bars: 20 μm (**a**,**d**), 100 μm (**b**,**i**), 50 μm (**b** – inset, **l**,**o**), 200 μm (**h**,**j**,**k**).

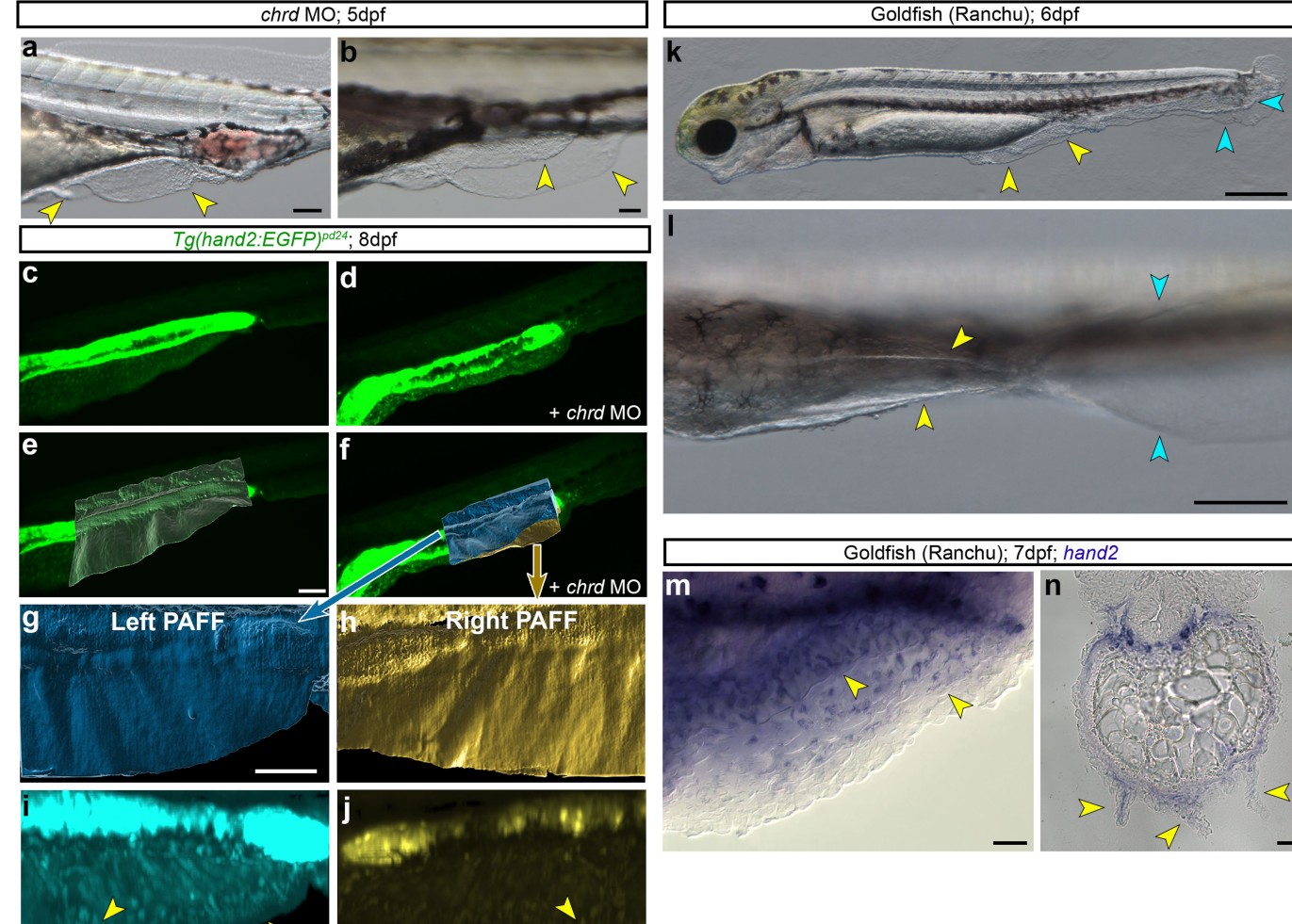

**Extended Data Fig. 8 | Reduced Chordin leads to paired duplication of PAFF. a**–**b**, Low (**a**) and high (**b**) power Nomarski images of duplicated PAFFs (yellow arrowheads) in 5 dpf larvae injected with *chrd* morpholino. **c**–**j**, Confocal images of PAFFs from 8 dpf *Tg(hand2:EGFP)* either uninjected (**c**,**e**) or injected with *chrd* morpholino (**d**,**f**,**g**–**j**). Confocal projections (**c**,**d**,**i**,**j**) and total surface renderings (**e**,**f**,**g**,**h**) highlight duplicated PAFFs (blue and yellow, **f**) compared to single PAFF in uninjected larvae (grey, **e**). Imaging of left (**g**) and right (**h**) duplicated PAFFs of *chrd* morphants indicated eGFP positive mesenchymal cells populated both fin folds (yellow arrowheads, **i**,**j**). **k**–**l**, Lateral (**k**) and

ventral (**l**) Nomarski images of the duplicated PAFFs of the Ranchu goldfish strain at 6 dpf. Duplicated PAFFs and ventral caudal fin folds indicated by yellow and cyan arrowheads, respectively. **m**–**n**, Lateral (**m**) and transverse (**n**) views of 7 dpf Ranchu larvae stained by in situ hybridisation for *hand2*. Expression in individual mesenchymal cells of the PAFF indicated by yellow arrowheads (**m**). Occurrence of three PAFFs with core *hand2* expression indicated by yellow arrowheads (**n**). Scale Bars: 100 μm (**a**,**e**,**g**), 50 μm (**b**), 500 μm (**k**), 200 μm (**l**), 20 μm (**m**,**n**).

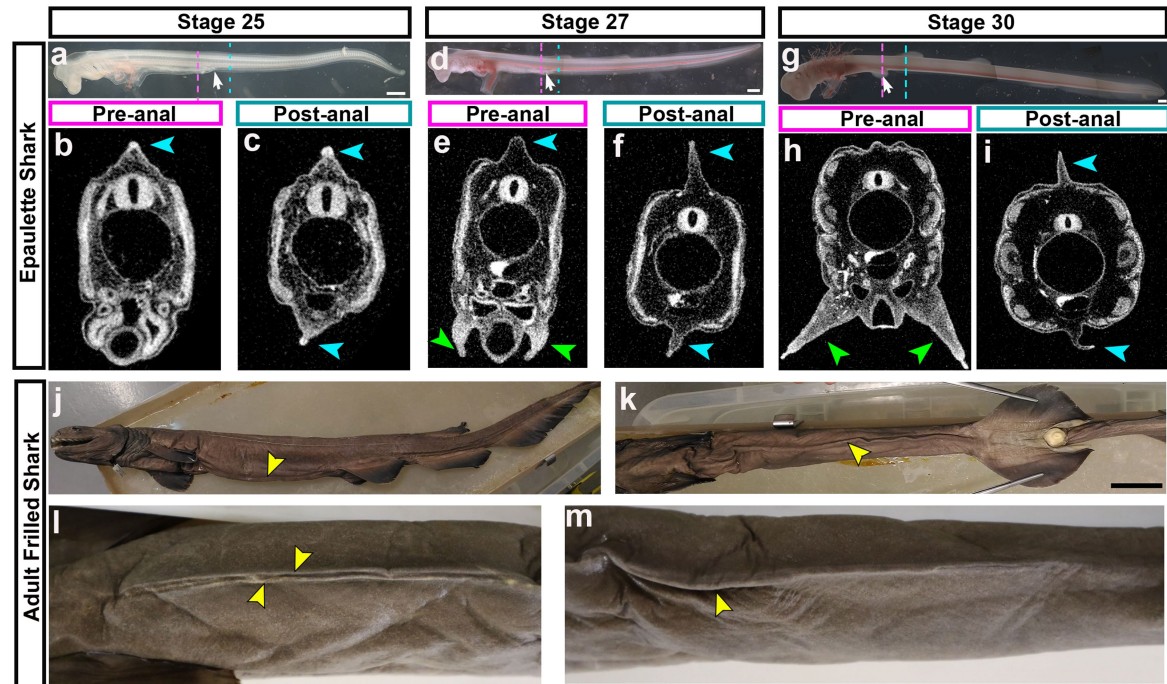

**Extended Data Fig. 9 | Variable presence of a pre-anal fin in sharks. a–i**, The Epaulette shark does not possess a PAFF. Whole mount (**a**,**d**,**g**) and virtual cross sections from microCT scans (**b**–**c**, **e**–**f**, **h**–**i**) of the Epaulette shark *Hemiscyllium ocellatum*, at the developmental stages of 25 (**a**–**c**), 27 (**d**–**f**), and 30 (**g**–**i**). Virtual cross sections for each stage show the region either anterior (Pre-anal; **b**,**e**,**h**), or posterior (Post-anal; **c**,**f**,**i**) to the developing cloaca (white arrow, **a**,**d**,**g**). The dashed lines (**a**,**d**,**g**) indicate approximate pre-anal (magenta) and post-anal (cyan) microCT section locations (separate specimens). The developing pelvic and median fin folds are indicated by green and cyan arrowheads respectively (**b**–**c**, **e**–**f**, **h**–**i**). **j**–**m**: The tropeic folds found in a pre-anal position of the adult frilled shark, *Chlamydoselachus anguineus*, shown in lateral (**j**) and ventral (**k**) views (yellow arrowheads) (adult male specimen, ZRC 54430 - LKC Natural History Museum, Singapore). **l**–**m**, Ventral view of another adult frilled shark exhibits partially paired tropeic folds (yellow arrowheads) in anterior half (**l**), which then merges to become singular (yellow arrowhead) in posterior half (**m**) (anterior: left, CSIRO H 7115-01, adult male 1310 mm TL, Tasmania, Australia). Scale Bars: 1 mm (a,d,g), 5 cm (**k**).

# Reporting Summary

## Statistics

For all statistical analyses, confirm that the following items are present in the figure legend, table legend, main text, or Methods section.

| n/a | Confirmed | |
|---|---|---|
| ☐ | ☒ | The exact sample size (*n*) for each experimental group/condition, given as a discrete number and unit of measurement |
| ☐ | ☒ | A statement on whether measurements were taken from distinct samples or whether the same sample was measured repeatedly |
| ☐ | ☒ | The statistical test(s) used AND whether they are one- or two-sided *Only common tests should be described solely by name; describe more complex techniques in the Methods section.* |
| ☒ | ☐ | A description of all covariates tested |
| ☒ | ☐ | A description of any assumptions or corrections, such as tests of normality and adjustment for multiple comparisons |
| ☐ | ☒ | A full description of the statistical parameters including central tendency (e.g. means) or other basic estimates (e.g. regression coefficient) AND variation (e.g. standard deviation) or associated estimates of uncertainty (e.g. confidence intervals) |
| ☐ | ☒ | For null hypothesis testing, the test statistic (e.g. *F*, *t*, *r*) with confidence intervals, effect sizes, degrees of freedom and *P* value noted *Give P values as exact values whenever suitable.* |
| ☒ | ☐ | For Bayesian analysis, information on the choice of priors and Markov chain Monte Carlo settings |
| ☒ | ☐ | For hierarchical and complex designs, identification of the appropriate level for tests and full reporting of outcomes |
| ☒ | ☐ | Estimates of effect sizes (e.g. Cohen's *d*, Pearson's *r*), indicating how they were calculated |

*Our web collection on statistics for biologists contains articles on many of the points above.*

## Software and code

Policy information about availability of computer code

| Data collection | Zeiss ZEN Blue 3.6 Fiji ImageJ ver. 1.52p IMARIS 9.9.1 (Oxford Instruments) Avizo 3D software (ThermoFisher Scientific) |
|---|---|
| Data analysis | GraphPad Prism 9 |

For manuscripts utilizing custom algorithms or software that are central to the research but not yet described in published literature, software must be made available to editors and reviewers. We strongly encourage code deposition in a community repository (e.g. GitHub). See the Nature Portfolio guidelines for submitting code & software for further information.

## Data

Policy information about availability of data

All manuscripts must include a data availability statement. This statement should provide the following information, where applicable:

- Accession codes, unique identifiers, or web links for publicly available datasets
- A description of any restrictions on data availability
- For clinical datasets or third party data, please ensure that the statement adheres to our policy

The authors confirm that all relevant data are provided in the results section of this paper, in its Extended Data files and in the Source Data file. The data for

measurements of fin size and cell number in Extended Data Figure 3m-o are also available in Figshare with the identifier https://doi.org/10.6084/m9.figshare.22269769.

## Human research participants

Policy information about <u>studies involving human research participants and Sex and Gender in Research.</u>

| | |
|---|---|
| Reporting on sex and gender | N/A |
| Population characteristics | N/A |
| Recruitment | N/A |
| Ethics oversight | N/A |

Note that full information on the approval of the study protocol must also be provided in the manuscript.

# Field-specific reporting

Please select the one below that is the best fit for your research. If you are not sure, read the appropriate sections before making your selection.

☐ Life sciences    ☐ Behavioural & social sciences    ☒ Ecological, evolutionary & environmental sciences

For a reference copy of the document with all sections, see nature.com/documents/nr-reporting-summary-flat.pdf

# Ecological, evolutionary & environmental sciences study design

All studies must disclose on these points even when the disclosure is negative.

| | |
|---|---|
| Study description | Expression analysis of tissue developmental origin across multiple species |
| Research sample | Danio rerio (WT strain: AB)<br>Oryzias latipes (WT strain: Cab)<br>Polyodon spathula<br>Xenopus tropicalis<br>Xenopus laevis<br>Petromyzon marinus<br>Carassius auratus (Ranchu strain) |
| Sampling strategy | n>=5 for all statistical analyses on fin size counts. All expression experiments were performed multiple times on different animal batches |
| Data collection | Data collected by microscopy |
| Timing and spatial scale | Developmentally relevant timepoints chosen |
| Data exclusions | Nil |
| Reproducibility | All expression experiments were performed multiple times on different animal batches. Findings were consistent across different repeat experiments |
| Randomization | Embryos selected at random for imaging |
| Blinding | NA |

Did the study involve field work?    ☐ Yes    ☒ No

# Reporting for specific materials, systems and methods

We require information from authors about some types of materials, experimental systems and methods used in many studies. Here, indicate whether each material, system or method listed is relevant to your study. If you are not sure if a list item applies to your research, read the appropriate section before selecting a response.

## Materials & experimental systems

| n/a | Involved in the study |
|-----|----------------------|
| ☐ | ☒ Antibodies |
| ☒ | ☐ Eukaryotic cell lines |
| ☒ | ☐ Palaeontology and archaeology |
| ☐ | ☒ Animals and other organisms |
| ☒ | ☐ Clinical data |
| ☒ | ☐ Dual use research of concern |

## Methods

| n/a | Involved in the study |
|-----|----------------------|
| ☒ | ☐ ChIP-seq |
| ☒ | ☐ Flow cytometry |
| ☒ | ☐ MRI-based neuroimaging |

## Antibodies

| | |
|---|---|
| Antibodies used | anti-Col2A1 (II-II6B3, DSHB)<br>anti-EGFP (TP401, Torrey Pines)<br>zns-5 (AB_10013796, ZIRC)<br>anti-SM22 alpha/Transgelin (ab14106, Abcam)<br>Alexa Fluor-488 Donkey anti-rabbit, Invitrogen, A21206<br>Alexa Fluor-647 Donkey anti-rabbit, Invitrogen, A31573<br>Alexa Fluor-546 Donkey anti-mouse, Invitrogen, A10036<br>Alexa Fluor-647 Donkey anti-mouse, Invitrogen, A31571 |
| Validation | See validation and associated references on the antibody supplier websites:<br>https://dshb.biology.uiowa.edu/II-II6B3<br>http://chemokine.com/Houston/rat&other/GFP.PDF<br>https://zfin.org/ZDB-ATB-081002-37<br>https://www.abcam.com/taglntransgelin-antibody-ab14106.html<br>https://www.thermofisher.com/antibody/secondary/query |

## Animals and other research organisms

Policy information about studies involving animals; ARRIVE guidelines recommended for reporting animal research, and Sex and Gender in Research

| | |
|---|---|
| Laboratory animals | Danio rerio (WT strain: AB)<br>Oryzias latipes (WT strain: Cab)<br>Polyodon spathula<br>Xenopus tropicalis<br>Xenopus laevis<br>Petromyzon marinus<br>Carassius auratus (Ranchu strain) |
| Wild animals | No wild animals were used in the study. |
| Reporting on sex | Most experiments were conducted during larval stages before sexual dimorphism occurs. Otherwise gender is reported for adults (in Extended Data Figure 9) |
| Field-collected samples | No field collected samples were used in the study |
| Ethics oversight | IMCB, A*STAR, Singapore (IACUC #140924)<br>Nanyang Technological University (IACUC #A18002)<br>CU Anschutz Medical Campus (protocol number 979)<br>National University of Singapore (IACUC #BR19-0120 and #BR22-1497)<br>James Madison University (IACUC #20-1601)<br>California Institute of Technology (IACUC #1436)<br>University of Manchester and the Home Office (PFDA14F2D)<br>Monash Animal Ethics Committee under license (#30347) |

Note that full information on the approval of the study protocol must also be provided in the manuscript.

