## [Peer Review File · Nature]

Manuscript Title: A lateral plate mesoderm derived median fin and the origin of paired fins

Reviewer Comments & Author Rebuttals

Reviewer Reports on the Initial Version:

Referee #1 (Remarks to the Author):

Review of Manuscript 2022-08-12320

T. J. Carney et al.

A lateral plate mesoderm derived median fin and the acquisition of paired fins

A. Summary of the key results

This manuscript documents expression patterns of genes specifically found in lateral plate mesoderm and its derivatives in the embryonic median pre-anal fin fold (PAFF) of zebrafish, medaka, paddlefish, lamprey, and *Xenopus*. These genes are not expressed in the embryonic median caudal fin fold of these species. Paraxial mesoderm is present in the caudal fin fold but not in the PAFF. The authors conclude that the PAFF is a lateral plate mesoderm-derived median fin fold. Because the paired pectoral and pelvic fins develop from lateral plate mesoderm and not paraxial mesoderm, the authors link this to the evolutionary origin of paired fins by bilateral duplication of the PAFF, which they show is possible by regulating expression of chordin. Relevant vertebrate fossils and comparative anatomy of the preanal fin fold are discussed.

B. Originality and significance: if not novel, please include reference

This is original and important information that contributes to longstanding questions about the evolutionary origin of paired fins in vertebrates.

C. Data & methodology: validity of approach, quality of data, quality of presentation

The descriptions and figures are clear, concise, and represent an excellent approach to questions about the evolution and development of fins.

D. Appropriate use of statistics and treatment of uncertainties

The statistics in support of Figure 2G are appropriate.

E. Conclusions: robustness, validity, reliability

The data speak for themselves and the conclusions about the role of the lateral plate mesoderm in the development of PAFF are justified.

The authors are careful not to speculate too much on the evolutionary significance of their results, but I would actually like to see their interpretation in Extended Data Figure 9 brought into the publication if at all possible. Such schematic diagrams have a long history in vertebrate biology, particularly in textbooks, and this would be an opportunity to highlight a new one.

F. Suggested improvements: experiments, data for possible revision

It would be interesting to study development of the PAFF in hagfishes, which are the only extant vertebrates that have a pre-anal fin as adults. This could be a logical extension for future research.

I also look forward to research showing in more detail how the fin developmental program was coopted from median to paired fins, a question touched upon in the manuscript.

I am a protagonist for using the dagger symbol (†) to flag extinct taxa. This is particularly important in a manuscript such as this one, which mentions both extinct and extant taxa. I would like to see this standardized for Nature journals because of their long history of publishing papers about fossils.

G. References: appropriate credit to previous work?

I appreciated the references to older papers including Cole, 1912 (although there seems to be a typographical error in that reference: XXXX).

However, the recently revived hypothesis that paired fins evolved from gill elements is not mentioned at all (e.g. Diogo, R. 2020; <https://anatomypubs.onlinelibrary.wiley.com/doi/10.1002/dvdy.192>). It is unclear whether or not this was a deliberate omission because the authors rule out that hypothesis as improbable, so it might be good to include it.

In reference 7, it should be M.V.H. Wilson; Reference 45 should be " of Scyliorhinus" not " ofScyliorhinus"; these presence of these minor errors suggests that the references should be carefully proofed.

H. Clarity and context: lucidity of abstract/summary, appropriateness of abstract, introduction and conclusions

The manuscript is clear and the context is excellent.

Thank you for the opportunity to read such an interesting contribution.

Sincerely,

Willy
William E. Bemis

Referee #2 (Remarks to the Author):

Review of Tzung et al 2022- Nature MS # 2022-08-12320.

This MS examines the developmental origin of the transient pre-anal fin fold (PAFF) in zebrafish and several other phylogenetically relevant taxa to support a hypothesis about the origin of paired appendages in vertebrates. The authors use multiple methods to test the embryonic lineage of the PFAA.

The lateral plate mesoderm (LPM) identity of mesenchymal cells within the PAFF is demonstrated by both positive and negative data.

- Labeling of paraxial mesoderm (somites, PM) with photoconvertible Kaede expression reveals PM cells for the post anal caudal fins and not the PFAA.
- Transgenics known to label LPM reveal cells in the PFA have a LPM origin, while no labeled cells are seen in the caudal fins.
- Mesenchymal identity of these cells in the PAFF is supported by Nomarski optics and specific molecular markers.
- ISH of the dHand transcription factor, which is known to label other LPM derivatives including pectoral and pelvic fins is also found in the PAFF but not in the caudal fins.
- Mutations known to specifically effect LPM derivative alter the morphology of the PAFF.
- Further evidence for the LPM source of the PAFF was confirmed by tracing lineage with the Tg(drl:creERT2:hsp701:Switch) transgenics- labeled cells found in paired appendages and the PAFF but not the caudal fin except for cells of a vascular type

The data is extensively illustrated and convincing. To place these data in a comparative framework, expression of dHand was used as a proxy for LPM in other species that also have a transient PAFF- Medaka, Polyodon, Petromyzon and Xenopus all show dHand expression in the PAFF, but not in caudal fins.

The text should distinguish between dorsal and ventral medial fins.

The authors state that the LPM origin of the PAFF is unexpected. Given that the LPM is critical for the closing of the coelom, it is hardly unexpected that the PRE- Anal fin is derived from that lineage. The caudal fins develop from the post anal tail bud, which is entirely PM, and thus these fins arise from PM.

The discussion focuses on the mesenchyme in the fin fold, but does not consider that a purely ectodermal fin fold may have predated the involvement of the LPM that then occurred through a persistence of the somatopleure, leading to the paired appendages.

Referee #3 (Remarks to the Author):

The authors present several new and important lines of evidence to support the possibility that vertebrate paired appendages arose via some kind of lateral duplication of ventral median fin folds. I am happy to recommend this article for publication in Nature with only minor revisions. My remarks are mainly about the evo-devo aspects, as my expertise hardly allows me to comment knowledgeably on the chosen molecular markers etc.

The relationship between median and paired fins and implications for the origin of the latter is a prominent question in vertebrate evolution, and this report contributes some much-needed insights. This manuscript is well written and concise.

The main results reported are

- 1) a teleost median ventral fin fold is populated by lateral plate mesodermal cells, like paired fins but different from the paraxial origin of mesoderm in anal and caudal fins;
- 2) this holds true also in non-zebra fishes including some phylogenetically informative taxa that indicate that LPM contribution to the PAFF is a vertebrate plesiomorphy, and so could have had a role in the evolutionary origin of paired appendages;
- 3) a morphogenetic manipulation can induce a lateral bifurcation of the PAFF with lateral plate-derived mesoderm; and
- 4) a naturally occurring novelty appears to recapitulate this kind of change. (I've always wanted to examine fantail goldfish for clues about the origin of paired fins.)

It is significant that this study focuses on a pre-anal ventral median fin fold: this structure lies within the antero-posterior domain of paired fin formation (indeed between the pelvic fin folds themselves), and can thus be distinguished from the fully developed ventral median fins.

The abstract is clearly written and presents the main results very well. The last sentence seems to offer support for the classical lateral fin fold hypothesis, whereas the strength of the presented research is in elucidating the developmental mechanisms underlying an evolutionary change. The old idea of an ancestral continuous fin fold along the ventrolateral flank from anus to gills which begat pairs of fins fore and aft really reflects the outdated conception of evolutionary transformation as changes from adult structure to adult structure. The research discussed in this report is the best kind of application of modern developmental biology to an evo-devo problem, and it provides evidence that a bilateral duplication of a median fin outgrowth may have arisen as anlagen for elaboration into paired fins.

Having said all that, the question is treated cogently in the last paragraph of the body of the text (lines 203-223).

Lines 58-59: I found the jump-cut from a general vertebrate development question to zebrafish in the lab a bit jarring. Perhaps a slightly gentler segue into the studies performed to address the question, which began with zebrafish work.

137-138: I always wondered about that pre-anal fin fold...

Lines 143-145: It is unmistakable that the authors understand the difference between inferring an ancestral condition from its shared derivation, but not all readers will necessarily be so clear on this point. Might it be possible to include Extended Data Fig. 9 in the main figures?

And although the meaning is clear, it is more precise to use 'origin' of paired fins in line 144 rather than 'development'.

189: What is 'representative' about Hemiscyllium? Or was it chosen to represent the elasmobranch clade?

221-223: 'novel evolutionary module' Does this mean new with the craniata? Or new because you just reported it?

Figures are excellent overall.

Fig. 1: in c, d, and e are the arrowheads indicating something particular?

Fig. 4: Does 'ventral confocal image' mean that c and d both in ventral view?
(And is d showing a duplicated anal opening?)

Fig. 9: The colors of the stars are only slightly different and may not reproduce well.

Referee #4 (Remarks to the Author):

This study seeks to clarify the emergence of paired appendages during evolution. I was recruited to comment specifically as an expert in zebrafish development. The data are generally of high quality, represent a significant step forward for our understanding of appendage development and evolution, and should be of broad interest. Nonetheless, there are some details that need to be addressed with respect to cellular and functional characterisation and the interpretation of some findings. The use of statistics appears appropriate.

- In order to confirm that the phenotypes observed in the pre-anal fin fold are cell autonomous, and given the proximity of the pre-anal fin fold to the *hand2* expression domain surrounding the yolk extension, it would be helpful to perform transplantation experiments of *hand2* loss of function cells onto wild type embryos (perhaps in the *Tg(hand2:EGFP)* line) and assess their contribution to the pre-anal fin fold.

- For me the major weakness in the current manuscript is in the use of *hand2* / *HandA* expression as proxy for an LPM origin across vertebrates. Although there is a correlation with the data from zebrafish, these experiments do not provide conclusive evidence for an LPM origin. As the zebrafish *drl* reporters are active in the equivalent LPM compartments across chordates, these findings should be validated in at least a couple of the other species examined – I assume medaka and *Xenopus* may be logistically the most straightforward and would significantly strengthen the conclusions made.

- *chrd* knockdown in zebrafish and analysis of the Rancho goldfish strain are intriguing, however again the use of *hand2* expression as a proxy of origin is not convincing.

Minor points:

- “Collectively, these results indicate that mesenchymal cells of the PAFF are functionally identical to those of the caudal median fin fold, yet have a distinct developmental origin.”

This may be true, however the analyses presented do not allow one to state this with such certainty. Rewording to “appear functionally indistinguishable” or something similar rather than “are functionally identical” would be more appropriate.

- It is unclear from the text why the *lyve1b:dsRed* zebrafish line was analyzed.

- The experiments using *drl:H2B-Dendra2* based lineage analyses are very elegant and convincing - in my opinion they deserve a place in the main figures.

- Whilst the difficulties of coordinating analyses across multiple species are recognised, the manuscript would benefit from more consistency in the in situ image analyses presented in Fig.3 and extended data figure 6.

Author Rebuttals to Initial Comments:

Replies to the Referees' comments on the original submission of "A lateral plate mesoderm derived median fin and the acquisition of paired fins" - (Nature 2022-08-12320)

First, we would like to thank all Referees for their time and effort in reviewing our manuscript. We have been strongly encouraged by their positive comments and valuable suggestions. We have taken their critical input to add to our previous submission and to improve on our manuscript. We have submitted our revised manuscript Figures 2 and 4 revised and have moved previous Extended Data Figure 9 to Main Figure 5. We have revised Extended Data Figures 3, 6 (now Extended Data Figure 7) and added a new Extended Data Figure 4. Our detailed replies to the specific requests and comments are as follows.

Referees' comments:

Referee #1:

Conclusions: robustness, validity, reliability

- *The authors are careful not to speculate too much on the evolutionary significance of their results, but I would actually like to see their interpretation in Extended Data Figure 9 brought into the publication if at all possible. Such schematic diagrams have a long history in vertebrate biology, particularly in textbooks, and this would be an opportunity to highlight a new one.*

Response:

In response to this suggestion from Referee 1 and a similar comment from Referee 3 we have now brought this in as Figure 5

Suggested improvements: experiments, data for possible revision

- *It would be interesting to study development of the PAFF in hagfishes, which are the only extant vertebrates that have a pre-anal fin as adults. This could be a logical extension for future research.*

Response:

We initially did look at 3 species of adult hagfish (*Eptatretus burgeri*, *Eptatretus okinoseanus* and *Eptatretus moki*) and found that, indeed, all have a well-formed adult pre-anal fin. However, this fin does appear different from the post-anal adult hagfish fins in that the pre-anal fin in hagfish contains no skeletal elements (see also Figure 1 of Ota, K. G *et al.* (2013). *J. Exp. Zool. B Mol. Dev. Evol.* 320, 129-139.).

We performed Hematoxylin and Eosin and Masson's Trichrome staining of transverse sections of *E. okinoseanus* and *E. moki* and saw abundant collagen fibers in the interstitial dermis within the fin. This ECM was embedded with dermal fibroblasts. We have included these images as a separate file for the Referee's interest.

We attempted to detect *HandA* (=hand2 orthologue) RNA in adult PAFF by RT-PCR, but we were unsuccessful in both *Myxine* and *Eptatretus* samples. This was, in hindsight, rather ambitious as there is little reason to expect a developmental gene such as *HandA* to remain this late to inform about lineage. We would need to obtain embryonic samples and attempt to detect LPM gene expression

(such as *HandA*) in nascent PAFF mesenchyme as we have done for other species, but the difficulty in obtaining hagfish embryos precludes such cell lineage analyses at this time point. We agree that this would be a logical extension for research and feel it would complement our lamprey data beautifully, but also add important evidence about the persistence of this LPM fin into adulthood.

- ***I also look forward to research showing in more detail how the fin developmental program was coopted from median to paired fins, a question touched upon in the manuscript.***

Response:

This is a direction we are indeed very keen to pursue as future next steps. We see expression of high-level regulators of mesenchyme differentiation programs in both pre-anal and post-anal unpaired fins as well as in embryonic pectoral fins, such as *pdgfra* and *msx* genes. We hope to obtain samples to perform broad phylogenetic transcriptomic analysis of the LPM and paraxial mesoderm, looking for shared regulators of the mesenchyme program. Ultimately, we hope to map co-option mechanisms to paraxial and lateral plate enhancers on such master regulator genes, and then use comparative genomics to define when these tissue specific enhancers arose in evolution with respect to PAFF and paired fin origin.

- ***I am a protagonist for using the dagger symbol (†) to flag extinct taxa. This is particularly important in a manuscript such as this one, which mentions both extinct and extant taxa. I would like to see this standardized for Nature journals because of their long history of publishing papers about fossils.***

Response:

We have labelled extinct taxa using the dagger symbol in Figure 5 (was Extended Data Figure 9).

References: appropriate credit to previous work?

- ***I appreciated the references to older papers including Cole, 1912 (although there seems to be a typographical error in that reference: XXXX).***

Response:

We have corrected the error which was due to us downloading the citation from Google Scholar and not checking it. Thanks so much for spotting this.

- ***However, the recently revived hypothesis that paired fins evolved from gill elements is not mentioned at all (e.g. Diogo, R. 2020; <https://anatomypubs.onlinelibrary.wiley.com/doi/10.1002/dvdy.192>). It is unclear whether or not this was a deliberate omission because the authors rule out that hypothesis as improbable, so it might be good to include it.***

Response:

That's very reasonable. We didn't omit it for any particular reason or obfuscation; just being focussed. But we agree that including it sets our work in the context of the larger historical debate for the reader. So, we have added mention of the original Gegenbaur translation and one of Andrew Gillis' papers in

the introduction paragraph. We find the model proposed by the Diogo paper combining both the gill-arch and lateral fin fold hypotheses to be very appealing. As it still requires evidence of lateral fin folds, our work does provide support for it, albeit indirectly. We have also included this reference in the introduction paragraph for the reader as it provides a comprehensive summary of the field. Thank you for pointing out this paper.

- *In reference 7, it should be M.V.H. Wilson; Reference 45 should be " of Scyliorhinus" not " ofScyliorhinus"; these presence of these minor errors suggests that the references should be carefully proofed.*

Response:

We have corrected the errors in the Reference 7 and 45 and tidied up other errors and duplications in other references.

We have also carefully proofread the References and will work with the Copy Editors to ensure they are all correctly formatted.

Referee #2:

- ***The text should distinguish between dorsal and ventral medial fins.***

Response:

Throughout, and especially in the figure legends, we have now specified which part of the caudal fin fold is analysed or imaged, where we were previously ambiguous or only one part is utilised. Where the entire caudal fin fold was analysed or imaged, we have left it unspecified as we intend to mean the whole caudal fin fold. We have also added a better description of the continuity of the single caudal (major) fin fold, from dorsal to caudal and then ventrally to the anus, to the second paragraph.

- ***The authors state that the LPM origin of the PAFF is unexpected. Given that the LPM is critical for the closing of the coelom, it is hardly unexpected that the PRE- Anal fin is derived from that lineage. The caudal fins develop from the post anal tail bud, which is entirely PM, and thus these fins arise from PM.***

Response:

Previous work in the Freitas paper (*Nature*, 2006) and our own previous work (Lee et al, *Development*, 2013) showed that median fins are Paraxial Mesoderm derived and paired fins are LPM. We always found it odd that our previous PM lineage work never labelled the PAFF. Our identification that this median fin derives from the LPM was indeed unexpected at the time given the literature on median fins. Nonetheless, the Referee's point highlights that our use of the term 'unexpected' in this sentence adds unnecessary subjectivity, and so we have removed this adjective.

- ***The discussion focuses on the mesenchyme in the fin fold, but does not consider that a purely ectodermal fin fold may have predated the involvement of the LPM that then occurred through a persistence of the somatopleure, leading to the paired appendages.***

Response:

The Referee raises an interesting and relevant point. Indeed, the larval amphioxus caudal fin is such an ectodermal only fin, however it is elongated by rather unique intracellular ciliary rootlets of epidermal cells. We predict that a purely ectodermal fin fold would be rather flimsy and may have been limited in size. Interestingly, following metamorphosis, the adult amphioxus fins grow further and are bolstered by mesoderm derived fin boxes and mesodermal strand cells which appear as infiltrating mesenchyme (Mansfield, J. H., & Holland, N. D. (2015). Amphioxus tails: Source and fate of larval fin rays and the metamorphic transition from an ectodermal to a predominantly mesodermal tail. *Acta Zool.*, 96(1), 117–125). How an LPM population acquires the competence to contribute to the fin fold could indeed be a function of LPM tissue context, as suggested by either the persistent somatopleure or lateral mesodermal divide models. These models could also account for subsequent regionalisation of the paired fins. We have added such considerations to the end of the discussion and clarified how our model interfaces with these concepts of LPM topology.

Referee #3:

- ***Lines 58-59: I found the jump-cut from a general vertebrate development question to zebrafish in the lab a bit jarring. Perhaps a slightly gentler segue into the studies performed to address the question, which began with zebrafish work.***

Response:

We have expanded the last half of the first paragraph to set the question in context better and with more accuracy. We hope this provides a smoother transition.

- ***Lines 143-145: It is unmistakable that the authors understand the difference between inferring an ancestral condition from its shared derivation, but not all readers will necessarily be so clear on this point. Might it be possible to include Extended Data Fig. 9 in the main figures?***

Response:

Referee 1 also had this request and so we have now brought this in as Figure 5.

- ***And although the meaning is clear, it is more precise to use 'origin' of paired fins in line 144 rather than 'development'.***

Response:

We have corrected it as suggested.

- ***189: What is 'representative' about Hemiscyllium? Or was it chosen to represent the elasmobranch clade?***

Response:

We indeed chose the Epaulette shark, *Hemiscyllium ocellatum* as a representative of the elasmobranch clade. It was readily available to us as a research colony. We realise the way we had written it was ambiguous and loaded, so we removed the term 'representative'.

- ***221-223: 'novel evolutionary module' Does this mean new with the craniata? Or new because you just reported it?***

Response:

We actually intended both meanings to be conveyed here and think that both are valid. We primarily propose that the PAFF was a novel invention in evolution, certainly in terms of its source of cellular contribution. But we also did mean to suggest that it is a novel concept for the field to integrate with current and future models, for the transition from unpaired to paired fins.

- ***Fig. 1: in c, d, and e are the arrowheads indicating something particular?***

Response:

The arrowheads in c, d, e indicate the PAFF, the post-anal (caudal), and the pectoral fin fold, respectively. Those fin folds show morphologically comparable mesenchymal cells. We have now included a statement in the figure legend to clarify this.

- ***Fig. 4: Does 'ventral confocal image' mean that c and d both in ventral view? (And is d showing a duplicated anal opening?)***

Response:

The images in c and d are both in ventral views, and we have re-written the figure legend to be more precise. (d) does indeed shows duplication of the anus. This is common with loss of Chordin and we have added explicit reference to this in the figure legend as well.

- ***Fig. 9: The colors of the stars are only slightly different and may not reproduce well.***

Response:

We've changed the colours of the stars to be more distinct in Figure 5 which is now in the main figures (was Extended Data Figure 9).

Referee #4:

- ***In order to confirm that the phenotypes observed in the pre-anal fin fold are cell autonomous, and given the proximity of the pre-anal fin fold to the hand2 expression domain surrounding the yolk extension, it would be helpful to perform transplantation experiments of hand2 loss of function cells onto wild type embryos (perhaps in the Tg(hand2:EGFP) line) and assess their contribution to the pre-anal fin fold***

Response:

We concur with the Referee that this is an interesting question, which we hope to pursue in the future (e.g., our ongoing work on mesothelial biology). We interpreted that as loss of Hand2 only affects the PAFF, and no other median fins, this supports an LPM origin of PAFF mesenchyme. As Hand2 is a transcription factor expressed in the PAFF mesenchyme, and the fact we see a specific PAFF mesenchyme phenotype upon loss of Hand2, we assumed that it was acting autonomously. However, it is true that these PAFF mesenchyme defects could be secondary to mesothelium defects in *hand2* mutants and thus not related to their LPM lineage. We think that this is unlikely. Firstly, the *hand2* mutant data is in line with the weight of independent experimental evidence we provide for the LPM origin of PAFF mesenchyme. Secondly many other mutants with disrupted mesothelium and yolk sac extensions have ostensibly normal PAFF morphology eg *wt1a/1b* mutants (*Dev. Biol.* **309**, 87-96 (2007)).

Nonetheless, to provide data on this, we have performed transplants of WT and *hand2*-deficient ET37 cells into WT hosts (Extended Data Figure 4). We performed over 600 transplants, but this was a very small target size and we hit this cell population very rarely. We were able to show that WT into WT transplants gave normal mesenchyme, but *hand2* MO into WT gave smaller clone sizes which showed altered cell morphology and limited migration. Moreover, it was even more difficult to get *hand2* MO transplants into the PAFF. This is evidence in favour of an autonomous role for Hand2 in PAFF mesenchyme, and argues that the PAFF defects in *hand2* mutants are indeed informing about lineage and supportive of our other lines of evidence.

- ***For me the major weakness in the current manuscript is in the use of hand2 / HandA expression as proxy for an LPM origin across vertebrates. Although there is a correlation with the data from zebrafish, these experiments do not provide conclusive evidence for an LPM origin. As the zebrafish drl reporters are active in the equivalent LPM compartments across chordates, these findings should be validated in at least a couple of the other species examined – I assume medaka and Xenopus may be logistically the most straightforward and would significantly strengthen the conclusions made.***

Response:

This is a highly valid point. We used the zebrafish *drl:EGFP* transgene to show that the PAFF mesenchyme in *Xenopus* and medaka came from *drl*-expressing domains (= LPM). This was shown in transient injections (*Xenopus* and medaka), but also through generating stable transgenics in medaka. As the *drl* promoter is active only in early LPM, the perdurance in PAFF mesenchyme is limited and the GFP levels are low. Nonetheless we consistently saw specific expression of GFP in the nascent PAFF mesenchyme of both *Xenopus* and medaka (8 cell across 4 transgenic medaka embryos). We additionally used DiI labelling of the end of the LPM in medaka to trace these LPM cells to the mesenchyme of medaka PAFF (27 cells in 9/9 labelled embryos). The results of these experiments have been included in Extended Data Figure 7. Of note, the medaka DiI experiments were directly

informed by our Dendra2-based lineage labelling of zebrafish LPM, providing the first position-level information of where the PAFF-seeding mesenchyme reside. Thus, we have used two independent methods to verify the interpretation of the *hand2* in situ approach in two species.

- ***chrd knockdown in zebrafish and analysis of the Rancho goldfish strain are intriguing, however again the use of hand2 expression as a proxy of origin is not convincing.***

Response:

We have performed *drl:CreERt2; hsp70:Switch* lineage tracing of LPM in *chordin* morphants to show that mesenchyme of both duplicated PAFFs are derived from the LPM. This is now included in Figure 4.

Our attempts to explore the possibility of microinjecting Rancho goldfish encountered several obstacles, both logistical and technical. We were unable to obtain quality breeding pairs from commercial breeders in the region. Furthermore, goldfish breeding can be problematic due to their seasonal breeding patterns. There is limited research on goldfish transgenesis with only a few published papers available, and none on Rancho. Consequently, the effectiveness and successful rate of transgenesis in goldfish, and Rancho in particular, remains uncertain. While transgenic lineage tracing would have been the ideal approach, it seems unlikely to be feasible within a reasonable timespan and without a significant investment of technical development. It is likely we would have to develop Cre-Lox reporter reagents in Rancho to do this.

We believe that the independent evidence we have gathered in other species, in particular in zebrafish, indicates *hand2 in situ* is highly likely reporting LPM lineage in Rancho. We see a highly specific pattern of *hand2* expression in PAFF mesenchyme, but not caudal fin mesenchyme, in both zebrafish and Rancho and confirm that this PAFF mesenchyme is indeed LPM derived in zebrafish using other approaches. It would be an exceptionally large coincidence that both Rancho and zebrafish have the identical specific restricted fin expression pattern of *hand2*, yet Rancho PAFF derives from another tissue. And indeed, we also see a similar restriction of *hand2* expression in other species and have shown for medaka and *Xenopus* that this also matches with lineage tracing approaches. Together these data make it extremely likely that *hand2* expression in Rancho PAFF indicates an LPM origin.

Minor points:

- ***“Collectively, these results indicate that mesenchymal cells of the PAFF are functionally identical to those of the caudal median fin fold, yet have a distinct developmental origin.” This may be true, however the analyses presented do not allow one to state this with such certainty. Rewording to “appear functionally indistinguishable” or something similar rather than “are functionally identical” would be more appropriate.***

Response:

For accuracy, we have now changed it to “mesenchymal cells of the PAFF show functional overlap with those of the caudal median fin fold”. We hope this wording is more precise.

- ***It is unclear from the text why the lyve1b:dsRed zebrafish line was analyzed.***

Response:

We observed previously that the promoter for the hyaluronan receptor, *lyve1b*, is active in the mesenchyme of both pre-anal and post-anal fins. This can be seen in the figures of (Okuda, K. S. et al. *Biol Open* 4, 1270-1280 (2015)). We have added this reference to the text for clarity. We used this to expand our comparison of the two populations of mesenchyme.

- ***The experiments using *drl:H2B-Dendra2* based lineage analyses are very elegant and convincing - in my opinion they deserve a place in the main figures.***

Response:

We much appreciate the Referee's positive take on these challenging experiments. We have included this experiment now in Figure 2, replacing the *hand2* mutant analysis which we have moved entirely to Extended Data Figure 3.

- ***Whilst the difficulties of coordinating analyses across multiple species are recognised, the manuscript would benefit from more consistency in the *in situ* image analyses presented in Fig.3 and extended data figure 6.***

Response:

We have tried to maintain consistency of images without playing with Photoshop image adjustment functions. Unfortunately, these varied appearances are a function of different PAFF fin sizes and mesenchyme numbers across the species, combined with the different *in situ* protocols and imaging set ups in our individual labs. The level of *hand2* expression in the PAFF mesenchyme was also variable in the different species, and we imaged the *in situ* signal in combination with Nomarski optics to show the context of these cells within the fin. This also generated variability in how these *in situ* images present. We have, however, prioritised that the results and conclusion from these *in situ* data are clearly shown in the panels.

Reviewer Reports on the First Revision:

Referee #1 (Remarks to the Author):

Thank you for your attention to comments that I raised about the previous version. All of these were answered and I appreciate the incorporation of Figure 5 as part of the paper. It adds a lot of context for readers.

Referee #2 (Remarks to the Author):

Response to revised MS.

In general, the authors have done a good job addressing the reviewers' comments, and the MS is ready for publication except for one point:

Perhaps due to the lovely new colors, and the placement of the summary cladogram in the main text (now fig. 5), I find the depiction of the lamprey very misleading- It looks very much like the adult has paired lateral fin folds of LPM origin. This of course is not the case.

It also reminded me of a similar cladogram published in 2011, Johanson Z (2011) Evolution of paired fins and the lateral somitic frontier. *Journal of vertebrate paleontology* 31 : 132 – 132

The new data on PAFF presented in Tzung et al., places the evolution of the LSF, a “novel evolutionary module” at a deeper node of the vertebrate tree. This is significant.

There is also a word missing in line 54.

Referee #3 (Remarks to the Author):

The authors have addressed the points I raised in the first round quite nicely. I have no more to add at this point.

Referee #4 (Remarks to the Author):

In the revised manuscript the authors present technically challenging experiments in multiple species to provide further experimental evidence for an LPM origin of PAFF mesenchyme, and for the use of Hand2 expression as a proxy for this origin across evolution. Collectively, they provide convincing evidence for a revised model for the emergence of paired appendages during evolution. The revised manuscript will be of broad interest to the readership of *Nature*.

Author Rebuttals to First Revision:

Replies to the Referees' comments on the revised submission of
"A lateral plate mesoderm derived median fin and the acquisition of paired fins" - (Nature 2022-08-12320)

Thank you to all Referees for looking at our revised version. Please find below outlined our corrections to the two referee comments on this version.

Referees' comments:

Referee #2:

- ***Perhaps due to the lovely new colors, and the placement of the summary cladogram in the main text (now fig. 5), I find the depiction of the lamprey very misleading- It looks very much like the adult has paired lateral fin folds of LPM origin. This of course is not the case. It also reminded me of a similar cladogram published in 2011, Johanson Z (2011) Evolution of paired fins and the lateral somitic frontier. Journal of vertebrate paleontology 31 : 132 – 132***

Response:

That illustration is of *Euphanerops* and was meant to be positioned at the end of the *Euphanerops* lineage on the cladogram. Its location close to the lamprey lineage was indeed confusing. We have realigned the cladogram and the constituent illustrations to be better aligned with their lineages. Thank you for that suggestion which has improved the clarity of the figure.

- ***There is also a word missing in line 54.***

Response:

This is corrected to "whilst a number of anatomists later invoked".